# Revisiting One-Shot Pruning with Scalable Second-Order Approximations

## Abstract

Pruning is a practical approach to mitigate the associated costs and environmental impact of deploying large neural networks (NNs). Early works, such as OBD (Le-Cun et al., 1989) and OBS (Hassibi & Stork, 1992), utilize the Hessian matrix to improve the trade-off between network complexity and performance, demonstrating that second-order information is valuable for pruning. However, the computation and storage of the Hessian matrix are infeasible for modern NNs, motivating the use of approximations. In this work, we revisit one-shot pruning at initialization (PaI) and examine scalable second-order approximations. We focus on unbiased estimators, such as the Empirical Fisher and the Hutchinson diagonal, that capture enough curvature information to improve the identification of structurally important parameters while keeping the linear computational overhead. Across extensive experiments on CIFAR-10/100 and TinyImagenet with ResNet and VGG architectures, we show that incorporating even coarse second-order information consistently improves pruning outcomes compared to first-order methods like SNIP and Hessian-vector product approaches like GraSP. We also analyze the problem of *layer collapse*, a significant limitation of *data-dependent* pruning methodologies, and demonstrate that simply updating the batch-norm statistics mitigates this problem. Notably, this warm-up phase substantially boosts the performance of the Hutchinson diagonal approximation in high sparsities, allowing it to surpass magnitude pruning after training (PaT), providing insight to possibly break through a long-standing wall for PaI methods (Frankle et al., 2020) and narrow the performance gap between PaI and PaT. Our results suggest that scalable second-order approximations effectively balance computational efficiency and accuracy, making them a valuable component of the pruning toolkit. Our code is made available to the public (*anonymized for review process*)[1].

## 1 Introduction

Rapid growth in computing power and data availability has enabled Neural Networks (NNs) to achieve remarkable progress in robotics (Soori et al., 2023), computer vision (Khan & Al-Habsi, 2020), natural language processing (Torfi et al., 2020), and related fields. However, this progress has come hand in hand with ever-growing model sizes, raising concerns about computational expense and energy consumption (Han et al., 2015). The prevailing "bigger is better" paradigm limits the deployment of large NNs in edge devices and resource-constrained settings (Cheng et al., 2024). Moreover, climate change awareness highlights the environmental impact of large-scale training and inference (Wu et al., 2022; Chien et al., 2023), making efficient learning an essential dimension for sustainable Artificial Intelligence (AI).

Model compression techniques, such as quantization (Dettmers et al., 2023), low-rank factorization (Denton et al., 2014), knowledge distillation (Xu et al., 2024), neural architecture search (Zhang et al., 2021), and neural network pruning (LeCun et al., 1989), aim to reduce model complexity while maintaining performance. Among these, pruning stands out because it can substantially reduce model size and computation workload with minimal performance drop. Cheng et al. (2024) provides a comprehensive taxonomy of pruning methods, focusing on three questions: (1) Does the pruning method achieve universal or specific acceleration? (2) At what stage of the training pipeline does the pruning occur? (3) Is the pruning criterion predefined or learned during training?

---

[1] https://anonymous.4open.science/r/revisiting-one-shot_pruning/

Most research has focused on pruning after training (PaT), given the benefit of working with converged models that provide reliable estimations of importance. Parameters with values close to optimal offer more information than randomly initialized ones (Kumar et al., 2024). PaT approaches range from simple magnitude pruning (Han et al., 2015) to more principled criteria that measure loss changes (Singh & Alistarh, 2020). Using the Hessian matrix has long been considered valuable in the PaT setting. The foundational works of Optimal Brain Damage (OBD) (LeCun et al., 1989) and Optimal Brain Surgeon (OBS) (Hassibi & Stork, 1992) introduced the idea of using second-derivative information to improve the trade-off between model complexity and performance.

Although effective, PaT comes at the expense of training a fully dense model, followed by parameter readjustment and/or retraining. This limitation raises interest for pruning at initialization (PaI), which aims to identify subnetworks before training. The milestone work of Frankle & Carbin (2018) established the existence of "winning tickets," sparse subnetworks that can match or even surpass the performance of their dense counterpart under the same parameter initialization. However, the iterative prune-retrain strategy to uncover them remains computationally heavy, motivating the search for more efficient search algorithms (Sreenivasan et al., 2022; You et al., 2022), or one-shot approaches to minimize computational overhead (Lee et al., 2018; Wang et al., 2020; Tanaka et al., 2020).

Incorporating second-derivative information directly at initialization seems to be a motivated step. However, there are reservations about relying on information from a randomly initialized model, given the high variance and the mismatch between the early and converged curvature information Liao & Mahoney (2021). Furthermore, the computation and storage of the Hessian matrix entries are infeasible for modern NNs. As a result, PaI research has favored first-order methods, such as SNIP (Lee et al., 2018), and gradient flow heuristics based on the Hessian-vector product (HVP), such as GraSP (Wang et al., 2020).

The findings of Gur-Ari et al. (2018) and Karakida et al. (2019) suggest that the curvature of a randomly initialized wide and overparameterized NN provides information of a small subspace where optimization takes place, which remains consistent during training. Motivated by these observations, we revisit the one-shot PaI setting to examine scalable estimators of the Hessian matrix for pruning. Specifically, we explore using diagonal approximations that allow for matrix-free computations. We study unbiased estimators of the Hessian diagonal, such as the Empirical Fisher Information Matrix (FIM) and Hutchinson's estimator, which reduce the overhead of second-order methods from quadratic to linear (Yao et al., 2021).

We also propose a method to mitigate *layer collapse*, a failure mode in *data-dependent* methods that disrupts the information flow between layers due to bottlenecks or complete layer removal (Tanaka et al., 2020). Gradient-based PaI methods often assign disproportionately low scores to wide layers, pruning them first and rendering the model untrainable (Kumar et al., 2024). We show that a simple warm-up phase to update batch normalization statistics induces better gradient estimation and prevents layer collapse. Notably, this adjustment substantially boosts the performance of the Hutchinson diagonal approximation in high sparsities ($\geq 90$), allowing it to surpass magnitude PaT, a long-standing barrier for PaI methods (Frankle et al., 2020).

Our experiments across CIFAR-10/100 and TinyImageNet with ResNet and VGG architectures demonstrate that incorporating coarse second-order information consistently improves pruning outcomes compared to long-standing one-shot PaI baselines like SNIP and GraSP. Our key contributions are summarized as follows: (1) we revisit one-shot PaI with scalable second-order approximations, focusing on empirical FIM and Hutchinson diagonal approximations for a balance between performance and computation overhead. (2) We show that updating batch normalization statistics provides a simple and effective fix to layer collapse and allows stable training. (3) We empirically demonstrate that principled pruning methods narrow the gap between PaI and PaT. And (4), our experiments show that even coarse curvature information outperforms long-standing PaI one-shot baselines, especially under high sparsity.

## 2 BACKGROUND & RELATED WORK

**Problem Setting**. Following the taxonomy in Wang et al. (2022), we focus on unstructured pruning. First, we create a static mask and then perform sparse training in a supervised learning setting. Given access to the training set $\mathcal{D} = \{(x_n, y_n)\}_{n=1}^{N}$, composed of tuples of input $x_n \in \mathcal{X}$ and output

$y_n \in \mathcal{Y}$, the goal is to learn a model parameterized by $w \in \mathbb{R}^d$ that maps $f : \mathcal{X} \to \mathcal{Y}$ by minimizing the objective function:

$$\mathcal{L}(w) = \frac{1}{N} \sum_{n=1}^{N} l(y_n, f(x_n; w)). \tag{1}$$

A binary mask $m \in \{0, 1\}^d$ is applied to induce sparsity, effectively reducing the parameter count of the model. We define the pruned model as $f(x_n; m \odot w)$, where $\odot$ denotes the Hadamard product between $m$ and the model weights $w$. The objective equation 1 then becomes:

$$\mathcal{L}(m \odot w) = \frac{1}{N} \sum_{n=1}^{N} l(y_n, f(x_n; m \odot w)). \tag{2}$$

We construct $m$ using an importance criterion and the target sparsity level. We denote $q \in \mathcal{Q}$, with $\mathcal{Q} = \{1, 2, \ldots, d\}$, as an index to refer to an element $w_q$ in the parameter vector $w$.

**The Hessian Matrix**. For a scalar-valued function that is twice differentiable, the matrix of second partial derivatives is called the Hessian:

$$H(w) := \nabla^2 \mathcal{L}(w) \in \mathbb{R}^{d \times d}. \tag{3}$$

It captures the local curvature of the loss landscape around $w$, indicating how rapidly the gradient changes in different directions. The direct computation and storage of the entire $d \times d$ matrix requires $\mathcal{O}(d^2)$ time and $\mathcal{O}(d^2)$ memory, making it unfeasible for modern NNs with millions, or even billions, of parameters.

Many second-order methods utilize an approximation of the Hessian entries to alleviate the computation, especially in optimization research. Hessian-free methods exploit the Hessian-vector (HVP) product, but might require many iterations or additional techniques for stability (Pearlmutter, 1994; Martens & Sutskever, 2011), adding computational overhead. Another approach is to approximate parts of the Hessian, such as the Generalized Gauss-Newton (GGN) method that uses the structure of the function to approximate a layer-wise block diagonal Hessian (Schraudolph, 2002). The Kronecker-factored Approximate Curvature method (KFAC) further reduces the GGN computation of each diagonal block, writing them as products of two smaller matrices (Martens & Grosse, 2015). Still, storing the Kronecker matrices can become prohibitively expensive for large models.

One can restrict the computation to the Hessian diagonal to minimize storage demand while preserving some curvature information. Although the exact calculation of the Hessian diagonal still has quadratic complexity, we can use stochastic methods to compute unbiased estimates (Elsayed et al., 2024). Martens et al. (2012) developed Curvature Propagation (CP) for an unbiased estimation of the Hessian diagonal. They can calculate a rank-1 approximation of the whole Hessian at the cost of two gradient evaluations. More recently, Yao et al. (2021) introduced the AdaHessian algorithm. It uses Hutchinson's unbiased estimator to approximate the diagonal of equation 3 using a multivariate random variable $z$ with a Rademacher distribution.

**The Fisher Information Matrix (FIM)**. The FIM has been employed as an approximation of the Hessian to increase computational speed (Vacar et al., 2011). It is defined as the expectation of the score function's second moment or the log-likelihood gradient. Based on the probabilistic concept that minimizing the loss function $l(y, f(x; w)$ is equivalent to maximizing the negative log-likelihood $-\log p(y \mid x, w)$, we can express the FIM in terms of the Hessian of the log-likelihood under regularity conditions (Schervish, 2012):

$$F(w) = -\mathbb{E}\left[\nabla^2 \log p(y \mid x, w)\right] = \mathbb{E}\left[\nabla^2 l(y, f(x; w))\right]. \tag{4}$$

Although the FIM approximation reduces the computation, the size of the FIM still requires $\mathcal{O}(d^2)$ memory. Soen & Sun (2024) elaborated on the trade-offs of approximating the FIM only by its diagonal to reduce its complexity to $\mathcal{O}(d)$. In practical settings, equation 4 is approximated using the empirical training distribution for an unbiased plug-in estimator, allowing the diagonal approximation to retain relevant geometric information while significantly reducing computational complexity, leading to the following formulation for the empirical FIM diagonal:

$$\text{diag}(\hat{F}) = \frac{1}{N} \sum_{n=1}^{N} \nabla l(y_n, f(x_n; w))^2. \tag{5}$$

This approximation has an intuitive interpretation: a given entry in $\text{diag}(\hat{F})$ corresponds to the average of the squared gradient of the model's output with respect to a parameter. The parameters influencing the model's output have larger entries, indicating higher importance.

**PaT Methods.** Early pruning methods established the deletion of parameters with low *saliency*, meaning their removal will have the least effect on performance. Hanson & Pratt (1988) compared different biases in the parameter search space and observed that parameters with larger magnitudes tend to be more significant. LeCun et al. (1989) went beyond the notion that "magnitude equals saliency" to present a more principled saliency measure. OBD employs a second-order Taylor series to measure the analytical change in loss by removing individual parameters under three assumptions: (1) the computationally expensive Hessian matrix $H$ is approximated using only its diagonal, which is more tractable; (2) assuming a converged model, the first-order term of the Taylor series is negligible; and (3) the local loss model is assumed quadratic, so the higher-order components are discarded. The saliency of parameter $q$ is defined as:

$$s_q = \tfrac{1}{2} w_q^2 H_{qq}. \tag{6}$$

A few years later, Hassibi et al. (Hassibi & Stork, 1992) revisited the approach with OBS. They highlighted the importance of a more complete representation of second-order information that includes off-diagonal elements and criticized the need to fine-tune the subnetwork. They redefined equation 6 as a constrained optimization:

$$\min_{q \in \mathcal{Q}} \min_{\delta w \in \mathbb{R}^d} \left\{ \tfrac{1}{2} \delta w^\top H \delta w \quad \text{s.t.} \quad e_q^\top \delta w + w_q = 0 \right\}, \tag{7}$$

where $e_q$ is a one-hot vector corresponding to $w_q$. Forming a Lagrangian from equation 7, they derived a general expression for saliency that includes equation 6 as a special case, and an expression to recalculate the magnitude of all parameters after removing a parameter $q$:

$$s_q = \frac{w_q^2}{2[H^{-1}]_{qq}}, \qquad \delta w = -\frac{w_q H^{-1} e_q}{[H^{-1}]_{qq}}. \tag{8}$$

OBS introduced a process to efficiently compute the inverse of the Hessian matrix $H^{-1}$ through the outer product approximation and the identity of the Woodbury matrix in its Kailath variant. Their main contributions rely on using a more accurate second-order estimation and $\delta w$ as a rescaling factor while pruning, making fine-tuning after pruning optional.

Theis et al. (Theis et al., 2018) proposed using empirical FIM to approximate Hessian in the PaT setting for the first time. As in equation 6, the first term of the Taylor expansion vanishes, but the FIM diagonal is used to approximate the Hessian diagonal. Their saliency metric is defined as:

$$s_q = \tfrac{1}{2} w_q^2 F_{qq}. \tag{9}$$

Using a similar approach for structured pruning, Liu et al. (2021) employed Fisher information to estimate the importance of channels identified by a layer grouping algorithm that exploits the network computation graph. Layers in the same group have the same pruning mask computed using the FIM diagonal. Inspired by OBS, Singh et al. (Singh & Alistarh, 2020) identified the challenges of computing and storing the inverse of the Hessian for large models and proposed a blockwise method to compute iteratively. The authors empirically showed the relationship between the empirical FIM inverse $\hat{F}^{-1}$ and the Hessian inverse $H^{-1}$, concluding that the first is a good approximation of the second as long as the application is scale invariant.

**PaI Methods.** When referring to methods that prune at initialization time, it is necessary to mention the *Lottery Ticket Hypothesis* introduced by Frankle et al. (Frankle & Carbin, 2018). Effective training depends heavily on the parameter initialization, hence the comparison to a lottery. Specific initializations enable the discovery of subnetworks that can be dramatically smaller than the dense original network and reach or exceed its performance. The authors supported their hypothesis with an iterative pruning approach that consistently found performant ResNet18 and VGG19 subnetworks with compression rates of $80-90\%$ for a classification task (CIFAR-10). This work's importance lies in demonstrating the existence of the winning tickets. However, they required a computationally expensive process, opening the question: If winning tickets exist, can we find them inexpensively?

Lee et al. (Lee et al., 2018) tried to answer that question with a single-shot network pruning method called SNIP, which measures the *connection sensitivity* as pruning criteria. They offered a different point of view on pruning, introducing an auxiliary vector of indicator variables $c \in \{0, 1\}^d$ that defines whether a connection is active or not, turning the pruning into a Hadamard product between $c$ and the parameters $w$. From that point of view, the method focuses on the influence of a connection on the loss function, named $\Delta \mathcal{L}_q$, approximating it as a directional derivative $g_q(w; \mathcal{D})$ that represents the effect of perturbation $c_q$.

$$\Delta \mathcal{L}_q(w) = \lim_{\epsilon \to 0} \frac{\mathcal{L}(c \odot w) - \mathcal{L}((c - \epsilon e_q) \odot w)}{\epsilon} \bigg|_{c=\mathbb{1}} = w_q \frac{\partial \mathcal{L}(w)}{\partial c_q} = g_q(w). \tag{10}$$

The intuition behind their proposed approach is that in PaI, the magnitude of the gradients with respect to the auxiliary vector $c$ indicates how each connection affects the loss, regardless of the direction. Allowing them to define their expression for the *connection sensitivity* as follows:

$$s_q = \frac{|g_q(c \odot w)|}{\sum_{k=1}^{m} |g_k(c \odot w)|}. \tag{11}$$

Wang et al. (2020) claimed that effective training requires preserving gradient flow through the model. They proposed a method heavily based on the neural tangent kernel (NTK) concept (Jacot et al., 2018), which provides the idea of how updates to a specific parameter affect others throughout the training process. By defining the gradient flow as the inner product $\nabla \mathcal{L}(w)^\top \nabla \mathcal{L}(w)$ they found a link to the NTK through its eigendecomposition, claiming that their method Gradient Signal Preservation (GraSP) helps to select the parameters that encourage the NTK to be large in the direction corresponding to the gradients of the output space. Similarly, they proposed a sensitivity metric that measures the response to a stimulus $\delta$:

$$S(\delta) = \Delta\mathcal{L}(w_0 + \delta) - \Delta\mathcal{L}(w_0) = 2\delta^\top H \nabla\mathcal{L}(w) + \mathcal{O}(||\delta||_2^2). \tag{12}$$

Tanaka et al. (2020) proposed a data-agnostic approach named SynFlow. Their pruning method is designed to identify sparse and trainable subnetworks within NNs at their initialization without training data. Previous methods require gradient information from the data and can inadvertently induce *layer collapse*, rendering the network untrainable due to information flow interruption. SynFlow addresses this by iteratively preserving the total "synaptic flow," or the cumulative strength of the connections, throughout the network during pruning. This approach ensures that essential pathways remain intact, maintaining the network's trainability.

## 3 METHODOLOGY

**Sensitivity Score.** Following OBD (LeCun et al., 1989), we start by approximating the objective equation 1 using a Taylor series. The perturbation $\delta w$ of the parameter vector will change $\mathcal{L}$ by

$$\delta\mathcal{L} = \mathcal{L}(w) - \mathcal{L}(w + \delta w) = \delta w^T \nabla\mathcal{L}(w) + \frac{1}{2}\delta w^T H \delta w + \mathcal{O}(||\delta w||^3). \tag{13}$$

Unlike OBD, the gradients are far from zero in the PaI setting, and neglecting the first-order term is not viable. Still, we operate assuming that the local error is quadratic to discard higher-order components. This approach assumes that individually deleting the parameters yields the same perturbation as removing them simultaneously.

As stated previously, it is not feasible to directly evaluate the quadratic term in equation 13 for large NNs. We aim to preserve linear complexity $O(d)$ to make the sensitivity score practical and minimize computational overhead. Therefore, we restrict the computation of the Hessian to diagonal approximations that capture enough curvature information. The equation 13 becomes:

$$\delta\mathcal{L} = \sum_{q \in \mathcal{Q}} \delta w_q \frac{\partial \mathcal{L}(w)}{\partial w_q} + \frac{1}{2} \sum_{q \in \mathcal{Q}} \delta w_q^2 H_{qq}. \tag{14}$$

Similar to Lee et al. (2018), we consider the term *sensitivity* more suitable than *saliency* in the PaI context. Induced perturbations at initialization will cause $\mathcal{L}$ to increase, decrease, or remain the same. Unlike inducing perturbations at convergence, where $\mathcal{L}$ only increases or stays the same. Consider the magnitude of equation 14. High values mean that the parameters $q$ significantly change the objective function (positive or negative) and should be preserved for the pruned model to learn. We define the sensitivity of the parameter $w_q$ as follows:

$$s_q = \left| w_q \frac{\partial \mathcal{L}(w)}{\partial w_q} + \frac{1}{2} w_q^2 H_{qq} \right|. \tag{15}$$

**Scalable approximation of the Hessian**. Firstly, we consider the Hutchinson approximation that AdaHessian (Yao et al., 2021) utilizes. We sample a random Rademacher vector $z \in \{-1, 1\}^d$ and compute the HVP $Hz$. In practice, computation of the HVP is performed via a single backpropagation trick without ever forming the full Hessian, keeping the cost on the order of a gradient pass. The elementwise product of $Hz$ and $z$, averaged over random draws, yields an unbiased estimator of the Hessian diagonal:

$$\mathrm{diag}(\hat{H}) = \mathbb{E}_z\big[(Hz) \odot z\big]. \tag{16}$$

With the incorporation of $\mathrm{diag}(\hat{H})$ into equation 15, we define Hutchinson-Taylor Sensitivity (HTS) as our first parameter importance score:

$$s_q = \left| w_q \frac{\partial \mathcal{L}(w)}{\partial w_q} + \frac{1}{2} w_q^2 \hat{H}_{qq} \right|. \tag{17}$$

Secondly, the equivalence between FIM and Hessian described in equation 4 only holds in convergence when the parameter vector $w$ maximizes the likelihood $\mathbb{E}[\nabla \log p(y \mid x, w)] = 0$. However, Karakida et al. (2019) showed that even at initialization, the FIM captures essential geometric properties of the parameter space. Some FIM eigenvalues are close to zero and indicate local flatness, while others are significantly large and induce substantial distortions in specific directions. Based on these claims, we test whether the signals from the empirical FIM diagonal are strong enough to identify important parameters. In addition, the FIM has the desirable property of being PSD by construction, ensuring a stable representation. With the incorporation of $\mathrm{diag}(\hat{F})$ into equation 15, we define Fisher-Taylor Sensitivity (FTS) as our second parameter importance score:

$$s_q = \left| w_q \frac{\partial \mathcal{L}(w)}{\partial w_q} + \frac{1}{2} w_q^2 \hat{F}_{qq} \right|. \tag{18}$$

**Baselines comparison and Taylor Series Ablation**. For comparison, we evaluate our proposed criteria against the following pruning methods: random, parameter magnitude, gradient norm (GN), SNIP Lee et al. (2018), and GraSP Wang et al. (2020). We also want to understand better the effect and capabilities of the evaluated Hessian approximations as pruning criteria, so we decompose the different elements of equation 15. The following methods employ the first-order component: magnitude, GN, and SNIP. Being the last reformulated by (Wang et al., 2020), showing that SNIP is equivalent to taking the magnitude of the first-order component:

$$s_q = \left| w_q \frac{\partial \mathcal{L}(w)}{\partial w_q} \right|.$$

From the second-order component, we evaluate two sensitivity criteria. First, we directly evaluate the diagonal approximations of the Hessian, referring to them as Hutchinson Diagonal (HD) and Fisher Diagonal (FD):

$$s_q = \hat{H}_{qq}, \qquad s_q = \hat{F}_{qq}. \tag{19}$$

Second, we evaluate the effect of using only the second-order term as in OBD (LeCun et al., 1989) or Fisher Pruning (Theis et al., 2018), referring to them as Hutchinson Pruning (HP) and Fisher Pruning (FP):

$$s_q = w_q^2 \hat{H}_{qq}, \qquad s_q = w_q^2 \hat{F}_{qq}. \tag{20}$$

**Pruning Mask**. Given a data set partition, we compute the vector $s$ that contains the sensitivity score $s_q$ for each parameter $q$. We generate the saliency scores throughout the training sets with a batch size of 1 for an accurate comparison. To create the pruning mask $m$, we define a percentile $p$ to narrow the subset containing the parameter index to retain:

$$\mathcal{R} = \{q \mid s_q \text{ is in the top } (1 - p) \text{ of scores}\}.$$

Using this subset $\mathcal{R}$, the elements of the binary mask $m$ are defined using the following rule: $m_q = 1$ if $q \in \mathcal{R}$ and $m_q = 0$ otherwise. We produce the pruned model $f(x; m \odot w_0)$ with the Hadamard product between the binary mask $m$ and the vector of the initial parameters of the model $w_0$, with the sparsity ratio defined as:

$$\text{sparsity} = \frac{1}{d} \sum_q m_q,$$

where $d$ is the total number of parameters of the dense model. This approach preserves only the parameters that have the most significant impact on the Taylor series approximation. Once the mask is applied, the pruned model is optimized utilizing stochastic gradient descent to minimize the objective function equation 2. It is important to mention that our pruning process skips the parameters from batch normalization and the output layers, as we consider them essential to enable learning and performing the designed task. We also skip bias parameters, as they are initialized as zero and are not informative in the PaI setting.

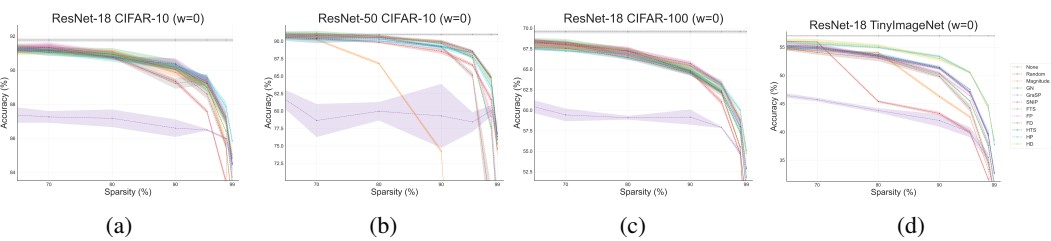

Figure 1: Test accuracy across sparsity levels for ResNet18/50 on CIFAR-10/100, and TinyImageNet under various PaI methods. The dashed gray line denotes the baseline accuracy without pruning.

## 4  RESULTS AND DISCUSSION

To evaluate the effectiveness of the different pruning criteria presented, we performed experiments on benchmarks commonly used in the pruning literature (Frankle et al., 2020), specifically CIFAR-10/100 (Krizhevsky et al., 2009) and TinyImageNet (Deng et al., 2009) datasets in ResNet (He et al., 2016) and VGG (Simonyan, 2014) architectures. We report all metrics across three initialization seeds, with training details in the Appendix A.

**Base Reference Case**. Given the low complexity of the task, we consider the CIFAR-10 classification with ResNet18 as our base reference case. Random pruning is the first trivial baseline that methods should surpass. As seen in Table A5 (Appendix E.1), random's performance is comparable to the 91.78 baseline accuracy up to 70% sparsity. After that point, the Plot 1a in Figure 1 shows a rapid performance degradation as we increase the sparsity. Although high sparsities require making trade-offs between potential efficiency improvements of sparsity and severe drops in accuracy, we focus our analysis on this regime, where the differences and limits of the methods are pronounced.

Magnitude pruning performs well up to 0.80 sparsity ratio, where it achieves the highest accuracy (91.10% ± 0.12). However, its performance decays rapidly in extreme sparsity, scoring lower than random in the sparsity ratio 90%. All methods (except GraSP) outperform random and magnitude pruning in sparsity ≥ 0.90, strengthening the argument that better-principled criteria are required to train pruned models efficiently. However, in the extreme sparsity regime, the HP criterion outperformed the rest of the methods, achieving 89.57% ± 0.08, 87.88% ± 0.27, and 85.86% ± 0.21 at sparsity ratios 95%, 98%, and 99%, respectively. These results suggest that even coarse second derivative information improves the trade-off between model complexity and performance.

**Increasing Model Complexity**. We evaluated the consistency of our results by switching the model in our reference case to ResNet50, a deeper and wider architecture model commonly used in large-scale vision tasks. Similarly, random's performance is comparable to the 90.97 baseline accuracy up to 70% sparsity (Table A7, Appendix E.2). The Plot 1b in Figure 1 further moves the balance in favor of principled criteria, as magnitude pruning suffers from layer collapse. Contrary to our reference case, SNIP and FTS are the top-performers in the extreme sparsity regime, with HD and HTS criteria constantly matching the top performer. Although there is a tendency for methods that incorporate first-order information in this experiment, three criteria incorporate coarse second-order information, suggesting robustness from the Hessian diagonal approximations.

**Increasing Task Complexity**. We also explore increasing the difficulty of the classification task by evaluating the CIFAR-100 and TinyImageNet datasets with ResNet18. As seen in the Plot 1c in Figure 1 (Table A10, Appendix F.1), the trend in CIFAR-100 is similar to that in the reference case. HP outperforms in the extreme sparsity regime, while all methods are comparable in low and moderate sparsities. The Plot 1d in Figure 1 (Table A13, Appendix G.1) shows the results for TinyImageNet. We can observe a considerable performance distance from the HD and HP criteria to the rest of the methods across the full sparsity range. These results suggest that second-order information that better approximates the Hessian outperforms methods that depend on first-order information as we increase the task complexity.

Our results, from the reference case to the high complexity ones, align with the observations of Yvinec et al. (2022), who note that a basic magnitude-based approach may remove low-magnitude parameters without considering their contribution to training. For a visual comparison and analysis

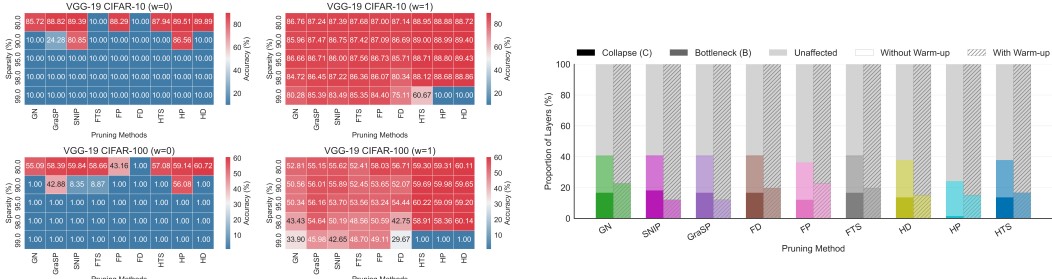

Figure 2: **Left:** Effect of a warm-up phase on pruning stability in VGG19 with CIFAR-10/100 at extreme sparsity ratios. Several methods exhibit systematic layer collapse without a warm-up (left), leading to near-random performance. A warm-up phase (right) largely prevents collapse and preserves accuracy across methods. Results for smaller sparsity ratios are reported in Appendix E.3. **Right:** Importance of batch normalization layers' statistics (BNS), preventing layer collapse. We studied the case of VGG19 pruning at 95% sparsity, which is the level that started showing layer collapse.

of parameter selection by the proposed criteria, we refer the reader to the Appendix I. Frankle et al. (2020) conclude that no single method is SOTA, since there is a network, a data set, and a sparsity combination where each pruning method reached the highest accuracy. We agree with their claim for low and moderate sparsity settings, where even random might be the top-performer. However, our results demonstrate that even coarse approximations of the Hessian improve the trade-off between sparsity and performance for extreme sparsities. Making scalable second-order approximations a valuable component of the pruning toolkit.

**Preventing Layer Collapse.** Table A8 (Appendix E.3) and Table A11 (Appendix F.2) show the performance of different pruning criteria evaluated in CIFAR-10 and CIFAR-100, respectively, with VGG19. All data-dependent methods (even magnitude pruning) suffer a drastic performance drop at higher sparsities. This behavior is consistent with the *layer collapse* phenomenon described by Tanaka et al. (Tanaka et al., 2020), where pruning removes entire layers (or most of their parameters), severely disrupting the flow of information and making the network untrainable. We propose a warm-up phase before forming the pruning mask, consisting of two steps: freeze all the model parameters and perform a complete pass over the dataset to update the batch-norm layers' statistics. This simple yet effective solution to mitigate layer collapse enables better gradient estimation at initialization, which is essential for all data-dependent criteria. The left side of Figure 2 shows that our proposed warm-up phase greatly alleviates layer collapse, with only a couple of criteria being unable to overcome it at 99% sparsity. The right side of Figure 2 presents a visual representation of the results in Table A4 (Appendix D). It is clearly observed that at the 0.95 sparsity ratio, the update of the batch-norm statistics reduces to 0% the number of collapsed layers and greatly reduces the percentage of bottlenecks, enabling the training of the pruned model.

**PaI vs. PaT comparison**. Frankle et al. (2020) strongly criticized PaI methodologies for consistently underperforming compared to magnitude PaT. However, their analysis did not assess how the PaI-designed criteria perform in the post-training setting. Table A14 in Appendix H.1 evaluates all criteria in the reference case under the same retraining protocol as Frankle et al. (2020). Our results show that all principled criteria outperform magnitude pruning in the PaT setting, with HP being the constant top-performer in high sparsities.

In Figure 3, we visualize the performance gain from the PaT setting over the PaI setting. Contrary to expectations, not all methods perform better in the PaT setting. GN and FD exhibit substantial performance drops in the PaT setting. At convergence, these methods might induce layer collapse. For magnitude pruning, as we increase the sparsity, it is clear that its advantage over PaI methodologies comes from training a fully dense model and retraining. In contrast, it is observed that HP is not only the constant top-performer in both settings, but it has the closest gap between them (1.6%).

For high sparsities, we evaluated whether the proposed warm-up phase has an improving effect in the PaI settings. Table A6 in Appendix E.1, shows that most methods increased their performance. Most notably, HD performance increased by 1.57%, 2.39%, and 1.96% for sparsity ratios 95%, 98%, and 99%, respectively. It is the only method that surpasses magnitude pruning in extreme sparsity. These

results solidify the value of scalable second-order information and provide insights to possibly break the long-standing wall for PaI methods without the training of a fully dense model.

**Computational Complexity**. Finally, we compare the pruning criteria on the basis of their complexity. Table A2 (Appendix B) details the practical and theoretical time complexity, space complexity, wall-clock runtime, and the relative percentage of training time of each method. Since we take the full training sets with a batch size of 1 for computing the scores, the wall-clock time reaches its limits for comparison. The proposed FTS and HTS effectively reduce the complexity of the Hessian matrix to $\mathcal{O}(d)$, and reduce the computational cost to just $\mathcal{O}(Nd)$, the same as first-order methods such as SNIP or GN. While the

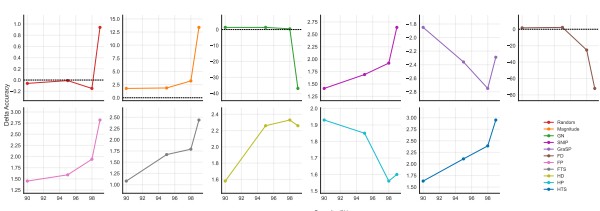

Figure 3: Delta between PaT and PaI test accuracy at extreme sparsities (0.90–0.99) for different pruning methods on ResNet18 with CIFAR-10. Each panel shows the behavior of one method across sparsity levels, with 0 marked by a dashed line. Results for smaller sparsity ratios are provided in Table A15, Appendix H.1.

mask computation of FTS barely contributes to the total training times, HTS incurs an outstandingly high wall-clock runtime given the requirement for $k$ random vectors $z$ from the Rademacher distribution. We defined $k = 10$ random vectors for each HVP pass for a better estimation, but the user can adjust to fewer if the number of passes is large enough. Also, in practical applications, the user can accelerate the wall-clock runtime with larger batch computations.

## 5 LIMITATIONS

Based on the findings of Gur-Ari et al. (2018) and Karakida et al. (2019), we operate under the assumption that the FIM and Hutchinson diagonals are good enough approximations of second-order information at initialization. The methods are designed to perform unstructured global pruning, limiting their applicability to attention-based architectures, which suffer from such strategies as pointed out in Cheng et al. (2024). The diagonal approximation assumes non-interaction between parameters at the second-order level score, leading to an inexpensive way of incorporating second-order information (see Appendix B). Additionally, we included an ablation to study the robustness of all data-dependent methods to noisy data when creating the pruning mask, finding that the results are consistent with our assumed noiseless setup in the reported results (see Appendix C).

## 6 CONCLUSION

This work presents scalable diagonal approximations of second-order information to perform one-shot pruning at initialization. We demonstrate the effectiveness of approximations in both PaI and PaT settings, especially in the extreme sparsity regime. The results show that HP and HD consistently outperform or match the SOTA PaI methods for different models, complexities of tasks, and sparsities. We demonstrated that a warm-up phase to update batch normalization statistics mitigates layer collapse when using data-dependent methods and improves the Hutchinson estimator.

Our work contributes to advancing efficient deep learning and resource-aware model deployment. We show that even coarse second-order methods reduce the performance gap between PaI and PaT, offering a step toward more efficient and theoretically grounded model compression techniques. Future work includes refining the approximations to capture off-diagonal interactions and extending our approach to structured pruning in attention-based architectures, as well as exploring the integration with other compression techniques, such as quantization.

## 7    ETHICS STATEMENT

This work does not involve human subjects, sensitive personal data, or applications with direct societal risk. The methods proposed focus on understanding and finding ways to improve neural network pruning efficiency at initialization, with the goal of reducing computational cost while maintaining model performance. By lowering the hardware requirements for training and inference, our approach contributes positively to the sustainability and accessibility of AI research. We do not anticipate ethical concerns beyond standard responsible research practices.

## 8    REPRODUCIBILITY STATEMENT

We have made all code, instructions, and experimental details available in an open-source repository (link provided in the abstract). The repository contains scripts for data preprocessing, model pruning at initialization, training, and evaluation, along with configuration files to replicate all reported results. Hyperparameters, experimental settings, and evaluation metrics are fully documented to ensure transparency. This facilitates independent verification of our findings and enables researchers to extend our work with minimal setup effort.

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

# APPENDIX

## A TRAINING AND TESTING DETAILS

We perform an 80:20 stratified split on the training sets to obtain a validation set with the same class distribution. Validation is performed after each training step, and the weights of the best-performing validation step (based on the top-1 accuracy) are used for the final evaluation of the test set. Table A1 shows the training parameters.

Table A1: Training configurations for various networks and datasets.

| Network | Dataset | Epochs | Batch | Opt. | Mom. | LR | Weight Decay | LR Drops | Drop Factor |
|---------|---------|--------|-------|------|------|-----|--------------|----------|-------------|
| ResNet-18 | CIFAR-10/100 | 160 | 512 | SGD | 0.9 | 0.01 | 1e-4 | 60, 120 | 0.2 |
| ResNet-50 | CIFAR-10 | 160 | 512 | SGD | 0.9 | 0.01 | 1e-4 | 60, 120 | 0.2 |
| VGG-19 | CIFAR-10/100 | 160 | 512 | SGD | 0.9 | 0.1 | 1e-4 | 60, 120 | 0.1 |
| ResNet-18 | TinyImageNet | 160 | 512 | SGD | 0.9 | 0.01 | 1e-4 | 30, 60, 80 | 0.1 |

All CIFAR-10, CIFAR-100, and TinyImageNet experiments were conducted in single GPUs NVIDIA RTX 5000.

# B  COMPUTATIONAL COMPLEXITY

For clarity, we will stick to the notation in the main body of the manuscript: $d$ denotes the number of parameters in the network, $B$ is the number of batches, and $N$ is the number of data samples. Some precision about the columns:

- **Practical Time Complexity** corresponds to how we compute it in the empirical setup, determined by the batch size used to create the mask and limited by available RAM/VRAM constraints. To compute the most accurate estimation of the Fisher Information Matrix (FIM), the batch size of the mask should be 1, which comes with the cost of more iterations when creating the mask ($B = N$).

- **Theoretical Time Complexity** indicates the worst-case scenario.

- **Space Complexity** indicates the storage requirements for estimating the metrics, considering that some require keeping intermediate results that will iteratively accumulate until the best metric estimation is obtained.

- **Wall-Clock Runtime** is the empirical time we measured for each method under similar conditions: the same task (ResNet-18 with CIFAR-10), the same type of GPU, uninterrupted computation, and a batch size of 1 (worst case).

- **Percentage of Training** Time refers to the portion of the training time the mask computation represents for each method.

Table A2: Comparison of pruning methods in terms of computational complexity and runtime characteristics.

| METHOD | PRACTICAL TIME COMPLEXITY | THEORETICAL TIME COMPLEXITY | SPACE COMPLEXITY | WALL-CLOCK RUNTIME (S) | TRAINING TIME % | DEPENDS ON DATA? | USES GRADIENTS? | HESSIAN USE |
|---|---|---|---|---|---|---|---|---|
| RANDOM | $\mathcal{O}(1)$ | $\mathcal{O}(1)$ | $\mathcal{O}(1)$ | 0.46 | 0.0002 | NO | NO | NONE |
| MAGNITUDE | $\mathcal{O}(d)$ | $\mathcal{O}(d)$ | $\mathcal{O}(d)$ | 0.07 | 0.00003 | NO | NO | NONE |
| GN | $\mathcal{O}(Bd)$ | $\mathcal{O}(Nd)$ | $\mathcal{O}(d)$ | 788.04 | 28.469 | YES | YES | NONE |
| SNIP | $\mathcal{O}(Bd)$ | $\mathcal{O}(Nd)$ | $\mathcal{O}(d)$ | 620.86 | 23.871 | YES | YES | NONE |
| GRASP | $\mathcal{O}(Bd^2)$ | $\mathcal{O}(Nd^2)$ | $\mathcal{O}(d)$ | 2880.69 | 59.265 | YES | YES | HESSIAN-VECTOR |
| **FD** | $\mathcal{O}(Bd)$ | $\mathcal{O}(Nd)$ | $\mathcal{O}(d)$ | 754.82 | 27.600 | YES | YES | DIAG. APPROX.* |
| **FP** | $\mathcal{O}(Bd)$ | $\mathcal{O}(Nd)$ | $\mathcal{O}(d)$ | 909.07 | 31.465 | YES | YES | DIAG. APPROX.* |
| **FTS** | $\mathcal{O}(Bd)$ | $\mathcal{O}(Nd)$ | $\mathcal{O}(d)$ | 986.83 | 33.262 | YES | YES | DIAG. APPROX.* |
| **HD** | $\mathcal{O}(Bd)$ | $\mathcal{O}(Nd)$ | $\mathcal{O}(d)$ | 15538.87 | 88.697 | YES | YES | DIAG. APPROX. |
| **HP** | $\mathcal{O}(Bd)$ | $\mathcal{O}(Nd)$ | $\mathcal{O}(d)$ | 15544.54 | 88.701 | YES | YES | DIAG. APPROX. |
| **HTS** | $\mathcal{O}(Bd)$ | $\mathcal{O}(Nd)$ | $\mathcal{O}(d)$ | 16469.52 | 89.268 | YES | YES | DIAG. APPROX. |
| OBD | $\mathcal{O}(Bd)$ | $\mathcal{O}(Nd)$ | $\mathcal{O}(d)$ | – | – | YES[a] | YES[b] | DIAG.[a] |
| OBS | $\mathcal{O}(Bd^2)$ | $\mathcal{O}(Nd^2)$ | $\mathcal{O}(d^2)$ | – | – | YES[a] | YES[c] | FULL INVERSE[d] |
| FULL HESSIAN | $\mathcal{O}(Bd^2)$ | $\mathcal{O}(Nd^2)$ | $\mathcal{O}(d^2)$ | – | – | YES | YES | FULL |

* FIM Approximation, [a] Post-Training, [b] For diagonal, [c] Outer products, [d] Kailath.

**Notes:**

1. F* methods use the empirical Fisher diagonal (computed from squared gradients on data batches), not the true Hessian. Practical runtime scales with the number of batches used for estimation.

2. H* methods use Hutchinson's stochastic diagonal approximation of the Hessian. Runtime scales linearly with the number of probe vectors (fixed here to [insert value, e.g., 10]). These require Hessian–vector products (second-order backprop), not just first-order gradients.

3. Depends on Data? indicates whether the method requires explicit forward/backward passes on input data to estimate its metric, not just parameter values.

4. Space Complexity: Entries such as $\mathcal{O}(d^2)$ (OBS, Full Hessian) are correct theoretically but infeasible in practice for modern networks, since storing the full Hessian or its inverse is memory-prohibitive.

5. Wall-clock times and training percentages for OBS and Full Hessian are omitted, as these methods are impractical at scale.

# C  NOISY MASK GENERATION

While pruning at initialization has shown promising results under standard conditions, real-world scenarios often involve some degree of noise or perturbation in the data. In practice, the data used to generate pruning masks may be affected by measurement errors, distributional shifts, or other sources of corruption. To evaluate the sensitivity of different data-dependent pruning methods to such perturbations, we conducted an experiment where Gaussian noise was added to the data loader used during mask generation. This setup allows us to assess the robustness of each method and to understand how performance is impacted when the pruning process is exposed to noisy inputs. This experiment considered our reference case (ResNet18 on CIFAR-10) with the same three seeds we used for our main results, using Gaussian noise with mean $0$ and std. $0.1$. We present the results in the following table; we only considered the data-dependent methods.

Table A3: Accuracy after generating a pruning mask, adding Gaussian noise for high-extreme sparsities for ResNet-18 and CIFAR-10 dataset.

| SPARSITY (%) | GN | SNIP | GRASP | FD | FP | FTS | HD | HP | HTS |
|---|---|---|---|---|---|---|---|---|---|
| 95 | 88.84 ± 0.22 | 89.14 ± 0.32 | 86.13 ± 0.23 | 89.49 ± 0.09 | 88.77 ± 0.13 | 89.14 ± 0.36 | **90.09 ± 0.22** | 89.07 ± 0.27 | **89.61 ± 0.82** |
| 98 | 85.95 ± 0.23 | 87.11 ± 0.35 | 85.93 ± 0.11 | 87.23 ± 0.29 | 85.84 ± 0.18 | 86.83 ± 0.09 | **88.15 ± 0.34** | 87.07 ± 0.32 | **88.04 ± 1.33** |
| 99 | 82.84 ± 0.56 | **84.53 ± 0.30** | **84.93 ± 0.10** | 84.47 ± 0.03 | 82.89 ± 0.60 | **84.28 ± 0.54** | **85.75 ± 0.29** | 84.59 ± 0.45 | **85.88 ± 1.06** |

The results reinforce the relevance of metrics relying on second-order information when facing data corruption. The HD criterion has the most consistent behavior across extreme sparsities, suggesting robustness that could be attributed to the sampling in Hutchinson's approximation process. It is noticeable that it was the only metric that boosted its performance from the setting without noise, in conjunction with GraSP, suggesting that there could be a benefit from building pruning masks with noise.

# D   STUDY ON BATCH-NORM STATISTICS

All models in our experiments use batch-norm (BN) layers; however, only VGG19 experiences layer collapse. We presumed that better statistics yield better gradient estimations, which is paramount when using data-based criteria in pruning. To validate this insight, we ran an experiment comparing the resulting pruning masks before and after updating the BN statistics.

Table A4: Importance of batch normalization layers' statistics (BNS) in preventing layer collapse. We studied the case of VGG-19 pruning at 95% sparsity, which is the level that started showing layer collapse. *C* indicates the percentage of layers that collapsed after pruning (100% of the layer is pruned), and *B* indicates the percentage leading to a bottleneck (80% of the layer is pruned).

| | GN | | SNIP | | GRASP | | FD | | FP | | FTS | | HD | | HP | | HTS | |
|---|---|---|---|---|---|---|---|---|---|---|---|---|---|---|---|---|---|---|
| BNS STATUS | C (%) | B (%) | C (%) | B (%) | C (%) | B (%) | C (%) | B (%) | C (%) | B (%) | C (%) | B (%) | C (%) | B (%) | C (%) | B (%) | C (%) | B (%) |
| WITHOUT WARM-UP | 16.67 | 24.20 | 18.18 | 22.73 | 16.67 | 24.24 | 16.67 | 24.24 | 12.12 | 24.24 | 16.67 | 24.24 | 13.64 | 24.24 | 1.52 | 22.73 | 13.64 | 24.24 |
| WITH WARM-UP | 0 | 22.73 | 0 | 12.12 | 0 | 12.12 | 0 | 19.70 | 0 | 22.73 | 0 | 19.70 | 0 | 15.15 | 0 | 15.15 | 0 | 16.67 |

The Table A4 compares the percentage of collapsed layers in VGG19 resulting from the pruning mask before and after updating only the BN statistics and leaving the initial parameters frozen using one complete pass through the data. Note that $\%C$ indicates the percentage of layers that collapsed after pruning (100% of the layer is pruned), and $\%B$ indicates the percentage leading to a bottleneck (80% of the layer is pruned). This insight allowed us to propose the warmup process to mitigate layer collapse.

# E   RESULTS CIFAR10

## E.1   RESNET18

Table A5: Performance of different pruning methods for CIFAR-10 on ResNet18 without warmup. The right side of the table presents our proposed criteria. Bold values highlight the top performer's mean accuracy and methods with matching performance (mean lies within the standard deviation of the best-performing method). Baseline, no pruning: $91.78 \pm 0.09$.

| SPARSITY (%) | RANDOM | MAGNITUDE | GN | SNIP | GRASP | FD | FP | FTS | HD | HP | HTS |
|---|---|---|---|---|---|---|---|---|---|---|---|
| 10 | **91.71 ± 0.21** | **91.72 ± 0.07** | 91.57 ± 0.15 | **91.72 ± 0.07** | 89.16 ± 0.05 | **91.87 ± 0.13** | **91.63 ± 0.21** | 91.53 ± 0.12 | 91.43 ± 0.20 | **91.59 ± 0.15** | 91.51 ± 0.07 |
| 20 | **91.63 ± 0.11** | **91.42 ± 0.12** | **91.51 ± 0.09** | **91.64 ± 0.16** | 88.69 ± 0.34 | **91.50 ± 0.12** | **91.65 ± 0.14** | **91.53 ± 0.15** | **91.57 ± 0.14** | **91.41 ± 0.18** | 91.29 ± 0.16 |
| 30 | **91.45 ± 0.18** | **91.61 ± 0.13** | **91.68 ± 0.20** | **91.65 ± 0.08** | 88.67 ± 0.26 | **91.65 ± 0.18** | **91.44 ± 0.27** | **91.49 ± 0.05** | 91.21 ± 0.07 | **91.32 ± 0.27** | **91.34 ± 0.14** |
| 40 | **91.59 ± 0.18** | 91.06 ± 0.16 | **91.61 ± 0.09** | **91.55 ± 0.08** | 88.24 ± 0.33 | **91.51 ± 0.05** | 91.38 ± 0.13 | **91.56 ± 0.28** | 91.24 ± 0.23 | **91.50 ± 0.12** | 91.39 ± 0.06 |
| 50 | **91.60 ± 0.15** | **91.32 ± 0.13** | **91.44 ± 0.13** | 91.22 ± 0.07 | 87.69 ± 0.15 | **91.30 ± 0.18** | **91.58 ± 0.16** | **91.46 ± 0.19** | **91.41 ± 0.20** | 91.31 ± 0.09 | 91.19 ± 0.14 |
| 60 | 91.10 ± 0.16 | 91.18 ± 0.16 | **91.59 ± 0.13** | 91.24 ± 0.04 | 87.48 ± 0.55 | 91.34 ± 0.07 | **91.35 ± 0.16** | **91.40 ± 0.11** | 91.00 ± 0.16 | 91.29 ± 0.08 | 91.26 ± 0.04 |
| 70 | **91.17 ± 0.04** | 91.07 ± 0.07 | **91.19 ± 0.17** | **91.33 ± 0.18** | 87.26 ± 0.34 | **91.34 ± 0.23** | **91.42 ± 0.23** | **91.21 ± 0.18** | 91.02 ± 0.08 | **91.10 ± 0.16** | **91.06 ± 0.21** |
| 80 | 90.78 ± 0.08 | **91.10 ± 0.12** | 90.95 ± 0.35 | 90.74 ± 0.10 | 87.18 ± 0.51 | 90.95 ± 0.11 | **91.08 ± 0.06** | 90.94 ± 0.22 | **90.90 ± 0.25** | **90.78 ± 0.27** | 90.81 ± 0.10 |
| 90 | 89.35 ± 0.13 | **89.88 ± 0.28** | **90.39 ± 0.23** | 90.36 ± 0.34 | 86.60 ± 0.51 | 90.04 ± 0.21 | 90.20 ± 0.08 | 89.22 ± 0.30 | **90.15 ± 0.14** | **90.25 ± 0.21** | **90.17 ± 0.16** |
| 95 | 87.59 ± 0.11 | 89.23 ± 0.19 | **89.00 ± 0.05** | 89.31 ± 0.17 | 86.50 ± 0.05 | 88.61 ± 0.28 | **89.50 ± 0.18** | **89.47 ± 0.32** | 89.09 ± 0.23 | **89.57 ± 0.08** | **89.23 ± 0.45** |
| 98 | 83.47 ± 0.20 | 85.70 ± 0.33 | 86.43 ± 0.05 | 87.26 ± 0.28 | 85.99 ± 0.08 | 85.61 ± 0.20 | 86.97 ± 0.22 | 87.24 ± 0.32 | 87.06 ± 0.35 | **87.88 ± 0.27** | 87.07 ± 0.20 |
| 99 | 78.28 ± 0.45 | 71.99 ± 0.28 | 83.47 ± 0.15 | 84.54 ± 0.04 | 84.56 ± 0.46 | 82.13 ± 0.28 | 83.74 ± 0.48 | 84.85 ± 0.18 | 85.05 ± 0.26 | **85.86 ± 0.21** | 84.48 ± 0.47 |

Table A5 shows the complete sparsity spectrum for ResNet18 with CIFAR-10. We highlight the model's resilience given that random pruning has a negligible drop in performance up to $0.7$ sparsity compared to the baseline. After this point, we observe a significant degradation for naive methods (random and magnitude). The proposed second-order-based criteria (in bold) demonstrate their robustness by consistently ranking among the top performers across sparsity ratios.

Table A6: Performance of different pruning methods for CIFAR-10 on ResNet18 after 1 warmup epoch on high to extreme sparsity. The right side of the table presents our proposed criteria. Bold values highlight the top performer's mean accuracy and methods with matching performance (mean lies within the standard deviation of the best-performing method). Baseline, no pruning: $91.78 \pm 0.09$.

| SPARSITY (%) | RANDOM | MAGNITUDE | GN | SNIP | GRASP | FD | FP | FTS | HD | HP | HTS |
|---|---|---|---|---|---|---|---|---|---|---|---|
| 80 | 90.73 ± 0.04 | 90.97 ± 0.24 | 91.38 ± 0.10 | 91.36 ± 0.03 | 83.82 ± 0.19 | **91.72 ± 0.13** | 91.32 ± 0.13 | 91.19 ± 0.09 | 91.28 ± 0.22 | 90.91 ± 0.24 | 90.85 ± 0.09 |
| 90 | 89.54 ± 0.10 | 90.10 ± 0.09 | 90.78 ± 0.08 | 90.58 ± 0.24 | 83.08 ± 1.14 | **91.06 ± 0.05** | 90.69 ± 0.08 | **90.79 ± 0.32** | **91.01 ± 0.16** | 90.52 ± 0.16 | 89.89 ± 0.03 |
| 95 | 87.43 ± 0.25 | 89.35 ± 0.13 | 89.77 ± 0.37 | 89.56 ± 0.22 | 84.18 ± 0.16 | 89.57 ± 0.62 | 89.62 ± 0.23 | 89.74 ± 0.10 | **90.66 ± 0.09** | 89.91 ± 0.15 | 88.90 ± 0.21 |
| 98 | 83.54 ± 0.03 | 86.69 ± 0.37 | 87.11 ± 0.09 | 87.47 ± 0.21 | 82.99 ± 0.54 | 86.30 ± 0.06 | 87.23 ± 0.28 | 87.85 ± 0.24 | **89.45 ± 0.11** | 88.07 ± 0.15 | 87.43 ± 0.10 |
| 99 | 78.22 ± 0.36 | 82.21 ± 0.35 | 83.82 ± 0.67 | 85.27 ± 0.31 | 82.58 ± 0.69 | 82.64 ± 0.26 | 84.85 ± 0.22 | 85.53 ± 0.26 | **87.01 ± 0.06** | 85.61 ± 0.14 | 84.72 ± 0.19 |

## E.2 RESNET50

Table A7: Performance of different pruning methods for CIFAR-10 on ResNet50 without warmup. The right side of the table presents our proposed criteria. Bold values highlight the top performer's mean accuracy and methods with matching performance (mean lies within the standard deviation of the best-performing method). Baseline, no pruning: $90.97 \pm 0.15$.

| SPARSITY (%) | RANDOM | MAGNITUDE | GN | SNIP | GRASP | FD | FP | FTS | HD | HP | HTS |
|---|---|---|---|---|---|---|---|---|---|---|---|
| 10 | 91.15 ± 0.08 | **91.23 ± 0.17** | **91.26 ± 0.21** | **91.43 ± 0.14** | 90.03 ± 0.60 | 91.17 ± 0.08 | 91.12 ± 0.14 | 91.00 ± 0.07 | 90.08 ± 0.27 | 90.29 ± 0.31 | 90.80 ± 0.37 |
| 20 | **91.03 ± 0.24** | **91.03 ± 0.12** | **91.27 ± 0.08** | **91.29 ± 0.19** | 88.53 ± 1.45 | **91.31 ± 0.25** | **91.06 ± 0.09** | **91.14 ± 0.03** | 90.24 ± 0.25 | 90.32 ± 0.16 | **91.03 ± 0.19** |
| 30 | **91.10 ± 0.16** | 90.75 ± 0.26 | **91.30 ± 0.10** | **91.26 ± 0.15** | 86.60 ± 1.81 | **91.38 ± 0.15** | **91.37 ± 0.22** | **91.23 ± 0.16** | 89.81 ± 0.31 | 90.40 ± 0.33 | 90.74 ± 0.18 |
| 40 | 90.92 ± 0.15 | 90.72 ± 0.16 | **91.24 ± 0.18** | 91.17 ± 0.13 | 87.05 ± 1.15 | **91.46 ± 0.14** | **91.22 ± 0.22** | 91.09 ± 0.13 | 90.43 ± 0.22 | 90.40 ± 0.25 | 90.93 ± 0.30 |
| 50 | 91.03 ± 0.11 | 90.56 ± 0.21 | **91.19 ± 0.06** | **91.12 ± 0.22** | 85.34 ± 1.53 | **91.26 ± 0.12** | 91.06 ± 0.02 | **91.37 ± 0.18** | 90.53 ± 0.22 | 90.42 ± 0.33 | 90.94 ± 0.22 |
| 60 | 90.66 ± 0.08 | 90.56 ± 0.09 | **91.23 ± 0.15** | 90.91 ± 0.21 | 84.63 ± 0.34 | **90.81 ± 0.27** | **91.02 ± 0.12** | **91.20 ± 0.31** | 90.89 ± 0.07 | 90.54 ± 0.16 | **90.79 ± 0.36** |
| 70 | 90.42 ± 0.24 | 90.22 ± 0.15 | **91.17 ± 0.30** | 90.60 ± 0.09 | 78.64 ± 2.36 | **90.82 ± 0.35** | **91.01 ± 0.09** | **90.76 ± 0.17** | 90.68 ± 0.23 | 90.24 ± 0.13 | **90.36 ± 0.64** |
| 80 | 89.85 ± 0.14 | 86.80 ± 0.19 | **90.71 ± 0.21** | **90.55 ± 0.08** | 79.96 ± 1.30 | **90.56 ± 0.18** | **90.68 ± 0.17** | **90.76 ± 0.17** | **90.60 ± 0.09** | 90.02 ± 0.07 | 90.31 ± 0.24 |
| 90 | 88.57 ± 0.36 | 74.26 ± 0.25 | **89.67 ± 0.13** | **89.97 ± 0.21** | 79.30 ± 4.57 | 89.17 ± 0.17 | **89.80 ± 0.15** | **89.87 ± 0.25** | **89.96 ± 0.17** | 89.20 ± 0.39 | 89.26 ± 0.17 |
| 95 | 86.59 ± 0.21 | 10.00 ± 0.00 | 87.65 ± 0.20 | **88.52 ± 0.25** | 78.43 ± 1.38 | 85.16 ± 0.43 | 87.97 ± 0.29 | **88.58 ± 0.17** | 88.31 ± 0.36 | 87.90 ± 0.17 | **88.34 ± 0.11** |
| 98 | 81.67 ± 0.51 | 10.00 ± 0.00 | 77.74 ± 0.57 | **84.81 ± 0.25** | 80.03 ± 0.78 | 65.62 ± 3.45 | 79.26 ± 0.87 | **84.79 ± 0.33** | 84.67 ± 0.68 | 83.60 ± 0.79 | **84.76 ± 0.18** |
| 99 | 74.57 ± 0.76 | 10.00 ± 0.00 | 65.63 ± 1.09 | **76.37 ± 0.44** | **76.88 ± 1.54** | 49.12 ± 0.36 | 67.05 ± 1.43 | **76.76 ± 0.30** | 75.25 ± 1.91 | 75.38 ± 1.80 | 75.95 ± 0.37 |

Table A7 shows the complete sparsity spectrum for ResNet50 with CIFAR-10. Compared to the smaller ResNet18 results, this deeper and wider architecture exhibits a faster performance drop as we increase the sparsity. Notably, the FD criterion is performant for low to sparsities, the GN criterion for moderate ones, and the SNIP criterion is for extreme ones. The proposed second-order criteria (bold) remain top performers across all sparsity levels, reinforcing the idea that there is not a silver bullet for PaI, and that criteria are complementary. However, we highlight that our proposed criteria manage to cover a larger part of the spectrum.

### E.3 VGG19

Table A8: Performance of different pruning methods for CIFAR-10 on VGG19. The right side of the table presents our proposed criteria. Bold values highlight the top performer's mean accuracy and methods with matching performance (mean lies within the standard deviation of the best-performing method). Baseline, no pruning: $89.21 \pm 0.22$.

| SPARSITY (%) | RANDOM | MAGNITUDE | GN | SNIP | GRASP | FD | FP | FTS | HD | HP | HTS |
|---|---|---|---|---|---|---|---|---|---|---|---|
| 10 | 88.40 ± 0.95 | 89.12 ± 0.55 | 90.14 ± 0.10 | 90.16 ± 0.18 | 87.81 ± 1.66 | 90.20 ± 0.29 | 90.21 ± 0.37 | **90.25 ± 0.38** | 89.50 ± 0.30 | 88.89 ± 0.74 | 90.04 ± 0.16 |
| 20 | 89.19 ± 0.22 | **89.65 ± 0.60** | 89.59 ± 0.69 | 90.06 ± 0.04 | 89.57 ± 0.34 | 89.91 ± 0.28 | **90.28 ± 0.55** | 89.80 ± 0.28 | 88.92 ± 0.11 | 88.30 ± 0.80 | **89.92 ± 0.39** |
| 30 | 88.93 ± 0.83 | 88.77 ± 1.07 | 90.23 ± 0.09 | 89.88 ± 0.59 | 89.14 ± 0.19 | 90.25 ± 0.09 | 89.97 ± 0.26 | **90.46 ± 0.41** | 89.97 ± 0.14 | 89.70 ± 0.26 | 89.82 ± 0.32 |
| 40 | 88.28 ± 1.08 | 89.38 ± 0.53 | 90.50 ± 0.23 | 89.79 ± 0.67 | 88.20 ± 0.31 | **90.51 ± 0.12** | 90.37 ± 0.24 | 90.23 ± 0.14 | 90.17 ± 0.15 | 90.09 ± 0.25 | 90.09 ± 0.23 |
| 50 | 88.96 ± 0.82 | 89.03 ± 0.59 | 90.46 ± 0.60 | 90.38 ± 0.25 | 88.67 ± 0.23 | 89.54 ± 0.86 | **90.47 ± 0.52** | 90.19 ± 0.31 | 90.05 ± 0.18 | 89.99 ± 0.11 | 89.97 ± 0.16 |
| 60 | 88.15 ± 0.68 | 89.47 ± 0.18 | **89.95 ± 0.30** | **90.32 ± 0.25** | 88.82 ± 0.32 | 90.02 ± 0.40 | 90.18 ± 0.33 | 90.14 ± 0.36 | 89.66 ± 0.79 | 89.51 ± 0.80 | 89.38 ± 1.12 |
| 70 | 88.02 ± 0.53 | **89.63 ± 0.44** | 89.69 ± 0.42 | 89.23 ± 0.19 | 89.62 ± 0.81 | 89.85 ± 0.08 | **90.01 ± 0.34** | 10.00 ± 0.00 | 89.85 ± 0.09 | 90.00 ± 0.07 | 89.61 ± 0.38 |
| 80 | 88.28 ± 0.34 | **89.62 ± 0.91** | 85.72 ± 0.63 | 89.39 ± 0.43 | 88.82 ± 0.14 | 10.00 ± 0.00 | 88.29 ± 0.11 | 10.00 ± 0.00 | **89.89 ± 0.54** | 89.51 ± 0.72 | 87.94 ± 0.09 |
| 90 | 85.82 ± 0.19 | **89.29 ± 0.79** | 10.00 ± 0.00 | 80.85 ± 0.62 | 24.28 ± 20.2 | 10.00 ± 0.00 | 10.00 ± 0.00 | 10.00 ± 0.00 | 10.00 ± 0.00 | 86.56 ± 0.53 | 10.00 ± 0.00 |
| 95 | **84.41 ± 0.05** | 10.00 ± 0.00 | 10.00 ± 0.00 | 10.00 ± 0.00 | 10.00 ± 0.00 | 10.00 ± 0.00 | 10.00 ± 0.00 | 10.00 ± 0.00 | 10.00 ± 0.00 | 10.00 ± 0.00 | 10.00 ± 0.00 |
| 98 | **80.04 ± 0.90** | 10.00 ± 0.00 | 10.00 ± 0.00 | 10.00 ± 0.00 | 10.00 ± 0.00 | 10.00 ± 0.00 | 10.00 ± 0.00 | 10.00 ± 0.00 | 10.00 ± 0.00 | 10.00 ± 0.00 | 10.00 ± 0.00 |
| 99 | **76.89 ± 0.26** | 10.00 ± 0.00 | 10.00 ± 0.00 | 10.00 ± 0.00 | 10.00 ± 0.00 | 10.00 ± 0.00 | 10.00 ± 0.00 | 10.00 ± 0.00 | 10.00 ± 0.00 | 10.00 ± 0.00 | 10.00 ± 0.00 |

Table A9: Performance of different pruning methods after warm-up phase for CIFAR-10 on VGG19. The right side of the table presents our proposed criteria. Bold values highlight the top performer's mean accuracy and methods with matching performance (mean lies within the standard deviation of the best-performing method). Baseline, no pruning: $89.21 \pm 0.22$.

| SPARSITY (%) | RANDOM | MAGNITUDE | GN | SNIP | GRASP | FD | FP | FTS | HD | HP | HTS |
|---|---|---|---|---|---|---|---|---|---|---|---|
| 80 | **88.73 ± 0.38** | **88.35 ± 0.54** | 86.76 ± 0.27 | 87.39 ± 0.66 | 87.24 ± 0.25 | 87.14 ± 0.45 | 87.00 ± 0.87 | 87.68 ± 0.33 | **88.72 ± 0.90** | **88.88 ± 0.95** | **88.95 ± 0.45** |
| 90 | 87.26 ± 0.42 | **88.62 ± 0.49** | 85.96 ± 0.75 | 86.75 ± 0.76 | 87.47 ± 0.33 | 86.69 ± 0.72 | 87.09 ± 0.31 | 87.42 ± 0.21 | **89.40 ± 0.46** | **88.99 ± 0.51** | **89.00 ± 0.24** |
| 95 | 85.47 ± 0.64 | 87.68 ± 0.49 | 86.66 ± 0.27 | 86.00 ± 1.10 | 86.71 ± 1.24 | 85.71 ± 1.35 | 86.73 ± 0.36 | 87.56 ± 0.62 | **89.43 ± 0.21** | 88.80 ± 0.80 | 88.71 ± 0.20 |
| 98 | 80.44 ± 0.30 | 86.61 ± 0.62 | 84.72 ± 1.69 | 87.22 ± 0.23 | 86.45 ± 0.64 | 80.34 ± 6.43 | 86.07 ± 0.39 | 86.36 ± 0.29 | **88.86 ± 0.54** | 88.68 ± 0.41 | 88.12 ± 0.75 |
| 99 | 77.24 ± 0.73 | **83.69 ± 1.36** | 80.28 ± 2.04 | 83.49 ± 1.77 | **85.39 ± 0.43** | 75.11 ± 7.80 | **84.40 ± 1.27** | 85.35 ± 1.05 | 10.00 ± 0.00 | 10.00 ± 0.00 | 60.67 ± 43.9 |

Table A8 shows the complete sparsity spectrum for VGG19 with CIFAR-10. We observe how performance drastically degrades for data-dependent methods when the sparsity increases, ultimately leading to layer collapse. As discussed in the Results section of the main body, we propose a warm-up phase that updates the batch norm statistics to prevent collapse and stabilize pruning performance. Table A9 demonstrates the effectiveness of this approach to mitigate layer collapse.

# F  RESULTS CIFAR100

## F.1  RESNET18

Table A10: Performance of different compression methods evaluated using ResNet18 on the CIFAR-100 dataset. Bold values highlight the top performer's mean accuracy and methods with matching performance (mean lies within the standard deviation of the best-performing method). The right side of the table presents our proposed criteria. Baseline, no pruning: $69.57 \pm 0.19$.

| SPARSITY (%) | RANDOM | MAGNITUDE | GN | SNIP | GRASP | FD | FP | FTS | HD | HP | HTS |
|---|---|---|---|---|---|---|---|---|---|---|---|
| 10 | 69.16 ± 0.11 | **69.37 ± 0.14** | **69.63 ± 0.34** | **69.42 ± 0.07** | 64.26 ± 0.27 | **69.66 ± 0.30** | 69.08 ± 0.21 | 69.16 ± 0.11 | 68.63 ± 0.53 | 69.03 ± 0.31 | **68.94 ± 0.56** |
| 20 | **69.16 ± 0.30** | **69.06 ± 0.24** | **69.19 ± 0.11** | **69.30 ± 0.08** | 63.28 ± 0.58 | **69.60 ± 0.30** | **69.35 ± 0.35** | **69.41 ± 0.43** | 68.39 ± 0.49 | 68.85 ± 0.26 | 68.60 ± 0.43 |
| 30 | **69.36 ± 0.18** | 68.58 ± 0.36 | **69.37 ± 0.13** | 68.82 ± 0.17 | 62.02 ± 0.43 | **69.24 ± 0.40** | 68.84 ± 0.13 | **68.80 ± 0.55** | 68.60 ± 0.46 | **69.20 ± 0.34** | 68.12 ± 0.42 |
| 40 | **69.41 ± 0.20** | 68.50 ± 0.29 | **69.16 ± 0.26** | 68.95 ± 0.19 | 61.18 ± 0.19 | **69.17 ± 0.16** | 68.88 ± 0.25 | **69.02 ± 0.21** | 68.01 ± 0.16 | 68.87 ± 0.13 | 68.15 ± 0.28 |
| 50 | **69.12 ± 0.46** | 68.17 ± 0.20 | **68.94 ± 0.20** | 68.63 ± 0.11 | 61.11 ± 0.40 | **69.13 ± 0.13** | 68.68 ± 0.12 | 68.71 ± 0.12 | **68.67 ± 0.53** | 68.37 ± 0.13 | 68.19 ± 0.27 |
| 60 | 68.66 ± 0.27 | 67.78 ± 0.35 | **68.77 ± 0.17** | 68.63 ± 0.42 | 61.40 ± 0.78 | **68.34 ± 0.43** | 67.98 ± 0.23 | 68.41 ± 0.14 | 68.10 ± 0.21 | 67.93 ± 0.25 | 67.71 ± 0.27 |
| 70 | 67.95 ± 0.43 | 67.51 ± 0.24 | **68.29 ± 0.39** | 68.08 ± 0.18 | 59.43 ± 0.76 | **68.03 ± 0.46** | **67.96 ± 0.15** | **68.29 ± 0.06** | 67.80 ± 0.40 | 67.21 ± 0.13 | **67.57 ± 0.35** |
| 80 | 67.26 ± 0.48 | 66.55 ± 0.19 | **67.20 ± 0.37** | **67.21 ± 0.38** | 59.08 ± 0.22 | 66.70 ± 0.05 | 67.05 ± 0.06 | **66.77 ± 0.65** | **67.32 ± 0.10** | 66.87 ± 0.04 | 66.42 ± 0.24 |
| 90 | 64.75 ± 0.16 | 64.48 ± 0.18 | 64.87 ± 0.27 | **65.70 ± 0.28** | 59.16 ± 0.91 | 64.74 ± 0.44 | **65.46 ± 0.30** | **65.41 ± 0.13** | **65.35 ± 0.47** | 64.89 ± 0.41 | 64.56 ± 0.33 |
| 95 | 61.01 ± 0.32 | 62.20 ± 0.06 | 62.20 ± 0.23 | **63.20 ± 0.20** | 57.91 ± 0.09 | 62.14 ± 0.42 | **63.22 ± 0.25** | **63.21 ± 0.47** | **62.87 ± 0.38** | **63.07 ± 0.41** | 62.36 ± 0.31 |
| 98 | 54.72 ± 0.22 | 55.44 ± 0.18 | 57.34 ± 0.31 | 58.83 ± 0.35 | 54.85 ± 0.35 | 55.57 ± 0.17 | 58.05 ± 0.18 | 58.59 ± 0.12 | 59.05 ± 0.51 | **60.05 ± 0.24** | 58.00 ± 0.49 |
| 99 | 45.62 ± 0.55 | 40.39 ± 0.36 | 50.46 ± 0.61 | 52.96 ± 0.12 | 49.13 ± 0.19 | 48.02 ± 0.32 | 49.98 ± 0.60 | 52.85 ± 0.24 | **54.73 ± 0.67** | **55.11 ± 0.46** | 51.86 ± 0.43 |

We tested the CIFAR-100 dataset to extend our evaluation with a higher-complexity task. Table A10 shows that although some of our proposed metrics remain top performers at high to extreme sparsities (FTS, HD, HP). Also, HD and HP criteria take a significant distance from SNIP (a traditional approach) in the extreme sparsity setting, suggesting that second-order information could capture information that is relevant to push the boundaries of PaI forward.

## F.2 VGG19

Table A11: Performance of different compression methods evaluated using VGG19 on the CIFAR-100 dataset. Bold values highlight the top performer's mean accuracy and methods with matching performance (mean lies within the standard deviation of the best-performing method). The right side of the table presents our proposed criteria. The right side of the table presents our proposed criteria. Baseline, no pruning: $58.96 \pm 2.30$.

| SPARSITY (%) | RANDOM | MAGNITUDE | GN | SNIP | GRASP | FD | FP | FTS | HD | HP | HTS |
|---|---|---|---|---|---|---|---|---|---|---|---|
| 10 | 60.31 ± 0.40 | 59.13 ± 1.29 | **61.93 ± 0.48** | 61.98 ± 0.29 | 59.32 ± 0.63 | 62.13 ± 0.61 | 60.45 ± 3.47 | 61.56 ± 1.04 | 57.14 ± 2.16 | 58.82 ± 1.48 | **60.71 ± 1.47** |
| 20 | 60.43 ± 1.14 | 59.27 ± 0.34 | 62.64 ± 0.21 | 62.68 ± 0.24 | 61.21 ± 0.41 | 63.04 ± 0.43 | 62.71 ± 1.02 | 62.24 ± 0.44 | 58.83 ± 1.37 | 58.62 ± 1.16 | 61.34 ± 0.62 |
| 30 | 58.32 ± 0.60 | 59.35 ± 1.43 | 62.61 ± 0.23 | 63.11 ± 0.35 | 59.30 ± 0.43 | 62.85 ± 0.42 | 61.43 ± 0.61 | 62.65 ± 0.54 | 61.27 ± 0.16 | 61.04 ± 0.50 | 61.90 ± 0.29 |
| 40 | 56.50 ± 3.20 | 60.04 ± 1.02 | 62.36 ± 0.02 | 62.39 ± 0.55 | 56.34 ± 1.49 | 62.38 ± 0.75 | 61.56 ± 1.25 | 62.67 ± 0.06 | 61.19 ± 0.50 | 60.59 ± 1.47 | 62.40 ± 0.19 |
| 50 | 58.47 ± 1.49 | 61.49 ± 1.22 | 62.02 ± 0.64 | 62.76 ± 0.50 | 54.43 ± 0.84 | 62.84 ± 0.33 | 62.25 ± 0.33 | 62.47 ± 0.42 | 61.00 ± 0.69 | 60.96 ± 0.83 | 59.76 ± 1.18 |
| 60 | 57.54 ± 0.74 | 61.50 ± 0.30 | 62.55 ± 0.13 | 63.08 ± 0.55 | 56.76 ± 0.69 | 62.40 ± 0.57 | 62.70 ± 0.63 | 62.17 ± 0.23 | 61.40 ± 0.59 | 61.45 ± 0.42 | 59.99 ± 3.42 |
| 70 | 57.63 ± 0.80 | 61.71 ± 0.25 | 60.85 ± 0.79 | 60.58 ± 0.39 | 57.76 ± 0.84 | 60.44 ± 0.34 | 60.92 ± 0.41 | 60.51 ± 1.67 | 61.45 ± 1.42 | 61.28 ± 0.18 | 60.42 ± 0.29 |
| 80 | 57.84 ± 0.57 | 61.89 ± 1.02 | 55.09 ± 0.49 | 59.84 ± 0.29 | 58.39 ± 0.74 | 1.00 ± 0.00 | 43.16 ± 1.02 | 58.66 ± 2.28 | 60.72 ± 0.38 | 59.14 ± 2.57 | 57.08 ± 0.50 |
| 90 | 58.41 ± 0.41 | 62.60 ± 0.91 | 1.00 ± 0.00 | 8.35 ± 10.39 | 42.88 ± 1.64 | 1.00 ± 0.00 | 1.00 ± 0.00 | 8.87 ± 11.13 | 1.00 ± 0.00 | 56.08 ± 0.72 | 1.00 ± 0.00 |
| 95 | 54.84 ± 1.08 | 1.00 ± 0.00 | 1.00 ± 0.00 | 1.00 ± 0.00 | 1.00 ± 0.00 | 1.00 ± 0.00 | 1.00 ± 0.00 | 1.00 ± 0.00 | 1.00 ± 0.00 | 1.00 ± 0.00 | 1.00 ± 0.00 |
| 98 | 50.21 ± 0.72 | 1.00 ± 0.00 | 1.00 ± 0.00 | 1.00 ± 0.00 | 1.00 ± 0.00 | 1.00 ± 0.00 | 1.00 ± 0.00 | 1.00 ± 0.00 | 1.00 ± 0.00 | 1.00 ± 0.00 | 1.00 ± 0.00 |
| 99 | 46.69 ± 0.45 | 1.00 ± 0.00 | 1.00 ± 0.00 | 1.00 ± 0.00 | 1.00 ± 0.00 | 1.00 ± 0.00 | 1.00 ± 0.00 | 1.00 ± 0.00 | 1.00 ± 0.00 | 1.00 ± 0.00 | 1.00 ± 0.00 |

Table A12: Performance of different compression methods evaluated after warm-up phase using VGG19 on the CIFAR-100 dataset. Bold values highlight the top performer's mean accuracy and methods with matching performance (mean lies within the standard deviation of the best-performing method). The right side of the table presents our proposed criteria. The right side of the table presents our proposed criteria. Baseline, no pruning: $58.96 \pm 2.30$.

| SPARSITY (%) | RANDOM | MAGNITUDE | GN | SNIP | GRASP | FD | FP | FTS | HD | HP | HTS |
|---|---|---|---|---|---|---|---|---|---|---|---|
| 80 | 60.39 ± 1.16 | 58.91 ± 0.41 | 52.81 ± 1.32 | 55.62 ± 2.27 | 55.15 ± 2.25 | 56.71 ± 0.31 | 58.03 ± 0.93 | 52.41 ± 3.07 | 60.11 ± 1.22 | 59.31 ± 2.79 | 59.30 ± 1.69 |
| 90 | 58.90 ± 0.98 | 60.95 ± 0.81 | 50.56 ± 4.59 | 55.89 ± 2.05 | 56.01 ± 1.58 | 52.07 ± 3.24 | 53.65 ± 0.57 | 52.45 ± 3.75 | 59.65 ± 2.16 | 59.98 ± 0.27 | 59.69 ± 1.32 |
| 95 | 56.10 ± 0.85 | 57.64 ± 2.63 | 50.34 ± 1.00 | 53.70 ± 3.60 | 56.16 ± 0.41 | 54.44 ± 1.38 | 53.24 ± 3.54 | 53.56 ± 1.26 | 59.20 ± 2.53 | 59.09 ± 3.86 | 60.22 ± 0.38 |
| 98 | 50.97 ± 0.40 | 54.66 ± 2.56 | 43.43 ± 5.32 | 50.19 ± 1.59 | 54.64 ± 1.50 | 42.75 ± 1.91 | 50.59 ± 3.39 | 48.56 ± 5.25 | 60.14 ± 0.28 | 58.36 ± 0.45 | 58.91 ± 0.14 |
| 99 | 46.52 ± 0.45 | 43.33 ± 5.83 | 33.90 ± 5.35 | 42.65 ± 5.32 | 45.98 ± 4.48 | 29.67 ± 8.49 | 49.11 ± 3.46 | 48.70 ± 2.59 | 1.00 ± 0.00 | 1.00 ± 0.00 | 1.00 ± 0.00 |

The results in VGG19 with CIFAR-100 exhibit a similar trend to those observed in CIFAR-10. Table A11 shows the occurrence of layer collapse in extreme sparsities when no warm-up is applied, leading to a significant drop in accuracy. Introducing a simple warm-up phase effectively resolves this issue as seen in Table A12, restoring the pruning performance in all evaluated criteria.

# G   Results TinyImageNet

## G.1   ResNet18

Table A13: Performance of different pruning methods for TinyImageNet on ResNet18. Bold values highlight the top performer's mean accuracy and methods with matching performance (mean lies within the standard deviation of the best-performing method). The right side of the table presents our proposed criteria. The right side of the table presents our proposed criteria. Baseline, no pruning: $57.00 \pm 0.11$.

| Sparsity (%) | Random | Magnitude | GN | SNIP | GraSP | FD | FP | FTS | HD | HP | HTS |
|---|---|---|---|---|---|---|---|---|---|---|---|
| 10 | $56.65 \pm 0.16$ | $\mathbf{56.88 \pm 0.22}$ | $56.48 \pm 0.19$ | $56.56 \pm 0.29$ | $53.94 \pm 0.34$ | $56.36 \pm 0.17$ | $56.84 \pm 0.09$ | $\mathbf{56.77 \pm 0.29}$ | $56.68 \pm 0.19$ | $\underline{\mathbf{57.25 \pm 0.23}}$ | $56.73 \pm 0.14$ |
| 20 | $\mathbf{56.43 \pm 0.25}$ | $\mathbf{56.52 \pm 0.19}$ | $\mathbf{56.56 \pm 0.18}$ | $56.31 \pm 0.09$ | $52.30 \pm 0.16$ | $\mathbf{56.50 \pm 0.34}$ | $\mathbf{56.35 \pm 0.14}$ | $\mathbf{56.49 \pm 0.12}$ | $\mathbf{56.58 \pm 0.40}$ | $\underline{\mathbf{57.06 \pm 0.60}}$ | $\mathbf{56.38 \pm 0.10}$ |
| 30 | $\mathbf{56.68 \pm 0.28}$ | $55.93 \pm 0.24$ | $56.31 \pm 0.14$ | $56.10 \pm 0.36$ | $50.97 \pm 0.53$ | $55.94 \pm 0.07$ | $56.18 \pm 0.29$ | $\mathbf{56.58 \pm 0.22}$ | $\mathbf{56.63 \pm 0.17}$ | $\underline{\mathbf{57.12 \pm 0.55}}$ | $56.34 \pm 0.22$ |
| 40 | $\mathbf{56.44 \pm 0.36}$ | $55.79 \pm 0.22$ | $\mathbf{56.22 \pm 0.15}$ | $\mathbf{56.06 \pm 0.27}$ | $50.05 \pm 0.27$ | $\mathbf{56.33 \pm 0.12}$ | $55.96 \pm 0.05$ | $55.94 \pm 0.04$ | $\underline{\mathbf{56.63 \pm 0.30}}$ | $\mathbf{56.62 \pm 0.21}$ | $55.94 \pm 0.27$ |
| 50 | $\mathbf{56.57 \pm 0.09}$ | $55.24 \pm 0.09$ | $\mathbf{56.21 \pm 0.30}$ | $55.86 \pm 0.27$ | $48.27 \pm 1.11$ | $55.98 \pm 0.04$ | $55.85 \pm 0.31$ | $55.95 \pm 0.21$ | $\mathbf{56.53 \pm 0.45}$ | $\underline{\mathbf{56.78 \pm 0.56}}$ | $\mathbf{56.02 \pm 0.40}$ |
| 60 | $\mathbf{56.07 \pm 0.11}$ | $54.94 \pm 0.50$ | $55.19 \pm 0.30$ | $55.53 \pm 0.27$ | $47.15 \pm 0.46$ | $55.41 \pm 0.18$ | $55.25 \pm 0.18$ | $55.52 \pm 0.11$ | $\underline{\mathbf{56.24 \pm 0.29}}$ | $\mathbf{56.06 \pm 0.45}$ | $55.16 \pm 0.47$ |
| 70 | $\underline{\mathbf{55.92 \pm 0.14}}$ | $54.33 \pm 0.12$ | $54.99 \pm 0.29$ | $55.17 \pm 0.39$ | $45.71 \pm 0.30$ | $54.06 \pm 0.40$ | $54.76 \pm 0.40$ | $54.91 \pm 0.18$ | $\mathbf{55.91 \pm 0.51}$ | $\mathbf{55.55 \pm 0.25}$ | $54.68 \pm 0.22$ |
| 80 | $45.39 \pm 0.18$ | $54.05 \pm 0.21$ | $53.42 \pm 0.32$ | $53.36 \pm 0.30$ | $43.79 \pm 0.39$ | $53.04 \pm 0.51$ | $53.30 \pm 0.34$ | $53.52 \pm 0.17$ | $\underline{\mathbf{55.20 \pm 0.30}}$ | $\mathbf{54.92 \pm 0.17}$ | $53.70 \pm 0.30$ |
| 90 | $43.27 \pm 0.28$ | $46.28 \pm 0.25$ | $50.34 \pm 0.26$ | $51.21 \pm 0.19$ | $42.03 \pm 1.08$ | $49.83 \pm 0.29$ | $50.37 \pm 0.33$ | $51.35 \pm 0.36$ | $\underline{\mathbf{52.98 \pm 0.38}}$ | $\mathbf{53.39 \pm 0.21}$ | $51.41 \pm 0.16$ |
| 95 | $39.96 \pm 0.34$ | $42.66 \pm 0.21$ | $45.48 \pm 0.16$ | $46.91 \pm 0.21$ | $39.94 \pm 0.86$ | $44.24 \pm 0.60$ | $46.41 \pm 0.13$ | $47.54 \pm 0.12$ | $\underline{\mathbf{50.55 \pm 0.18}}$ | $\mathbf{50.52 \pm 0.19}$ | $47.13 \pm 0.20$ |
| 98 | $31.69 \pm 0.22$ | $33.34 \pm 0.10$ | $37.63 \pm 0.45$ | $39.43 \pm 0.28$ | $35.51 \pm 0.75$ | $34.53 \pm 0.67$ | $38.16 \pm 0.34$ | $39.33 \pm 0.07$ | $44.30 \pm 0.35$ | $\underline{\mathbf{44.75 \pm 0.07}}$ | $39.71 \pm 0.62$ |
| 99 | $24.11 \pm 0.13$ | $24.41 \pm 0.14$ | $30.56 \pm 0.73$ | $32.71 \pm 0.52$ | $28.41 \pm 0.85$ | $28.94 \pm 0.36$ | $31.05 \pm 0.14$ | $32.86 \pm 0.36$ | $\underline{\mathbf{38.55 \pm 0.27}}$ | $37.77 \pm 0.47$ | $32.75 \pm 0.26$ |

We tested the TinyImageNet dataset to extend our evaluation with a higher-complexity task. Table A13 shows that the metrics based on Hutchinson approximation show a clear dominance in a more complex task, being HD and HP the sole best performing metrics at the most extreme sparsities we tried.

# H  PRUNE AFTER TRAINING CIFAR-10

## H.1  RESNET18

Table A14: Pruning After Training performance of different methods evaluated using ResNet18 on CIFAR10 after retraining. Bold values highlight the top performer's mean accuracy and methods with matching performance (mean lies within the standard deviation of the best-performing method). The right side of the table presents our proposed criteria. The right side of the table presents our proposed criteria. Baseline, no pruning: $91.83 \pm 0.15$.

| SPARSITY (%) | RANDOM | MAGNITUDE | GN | SNIP | GRASP | FD | FP | FTS | HD | HP | HTS |
|---|---|---|---|---|---|---|---|---|---|---|---|
| 10 | **92.47 ± 0.18** | 92.38 ± 0.09 | 92.29 ± 0.17 | 92.29 ± 0.11 | 88.89 ± 0.15 | **92.37 ± 0.12** | 92.36 ± 0.11 | 92.34 ± 0.04 | 92.27 ± 0.20 | 92.36 ± 0.14 | 92.34 ± 0.09 |
| 20 | **92.44 ± 0.25** | 92.26 ± 0.16 | **92.43 ± 0.02** | **92.50 ± 0.05** | 86.90 ± 0.54 | **92.41 ± 0.17** | 92.37 ± 0.07 | **92.51 ± 0.07** | 92.30 ± 0.11 | 92.31 ± 0.07 | **92.46 ± 0.13** |
| 30 | **92.43 ± 0.09** | 92.05 ± 0.17 | **92.47 ± 0.12** | 92.37 ± 0.06 | 86.19 ± 0.98 | **92.51 ± 0.07** | 92.33 ± 0.15 | 92.33 ± 0.08 | 92.31 ± 0.08 | **92.35 ± 0.18** | 92.37 ± 0.15 |
| 40 | **92.51 ± 0.10** | 92.20 ± 0.16 | **92.39 ± 0.19** | 92.17 ± 0.07 | 85.31 ± 0.92 | **92.35 ± 0.13** | 92.22 ± 0.07 | **92.46 ± 0.10** | 92.22 ± 0.17 | 92.21 ± 0.16 | **92.44 ± 0.16** |
| 50 | **92.35 ± 0.13** | 92.10 ± 0.12 | **92.44 ± 0.10** | 92.23 ± 0.10 | 85.99 ± 0.30 | 92.23 ± 0.09 | **92.34 ± 0.11** | **92.40 ± 0.17** | 92.11 ± 0.08 | **92.32 ± 0.22** | 92.23 ± 0.20 |
| 60 | 92.06 ± 0.24 | 92.06 ± 0.03 | **92.46 ± 0.07** | 92.13 ± 0.12 | 85.89 ± 0.28 | 92.15 ± 0.09 | 92.19 ± 0.02 | 92.17 ± 0.16 | 92.15 ± 0.10 | **92.37 ± 0.26** | 92.29 ± 0.08 |
| 70 | 91.74 ± 0.01 | 92.05 ± 0.05 | **92.32 ± 0.20** | 92.29 ± 0.20 | 84.96 ± 0.29 | **92.27 ± 0.14** | 92.15 ± 0.11 | **92.35 ± 0.05** | 92.13 ± 0.10 | **92.28 ± 0.21** | 92.22 ± 0.17 |
| 80 | 90.66 ± 0.11 | 91.96 ± 0.07 | **92.36 ± 0.07** | 92.02 ± 0.08 | 84.47 ± 0.56 | **92.32 ± 0.10** | 91.93 ± 0.04 | 92.02 ± 0.10 | 91.85 ± 0.10 | **92.22 ± 0.11** | 92.13 ± 0.01 |
| 90 | 89.29 ± 0.35 | 91.65 ± 0.21 | 91.74 ± 0.16 | 91.77 ± 0.24 | 84.75 ± 0.09 | 91.78 ± 0.13 | 91.65 ± 0.12 | 91.66 ± 0.07 | 91.73 ± 0.17 | **92.18 ± 0.17** | 91.80 ± 0.13 |
| 95 | 87.58 ± 0.20 | **91.09 ± 0.17** | 90.34 ± 0.65 | 91.00 ± 0.08 | 84.14 ± 0.56 | 90.89 ± 0.09 | 91.09 ± 0.16 | **91.14 ± 0.21** | **91.35 ± 0.12** | **91.42 ± 0.16** | 91.34 ± 0.20 |
| 98 | 83.32 ± 0.30 | **88.90 ± 0.18** | 86.88 ± 1.01 | **89.18 ± 0.19** | 83.24 ± 0.32 | 60.28 ± 35.56 | **88.91 ± 0.18** | 89.03 ± 0.03 | 89.39 ± 0.19 | 89.44 ± 0.14 | **89.46 ± 0.43** |
| 99 | 79.22 ± 0.02 | 85.38 ± 0.26 | 46.55 ± 27.39 | **87.18 ± 0.22** | 82.27 ± 0.57 | 10.00 ± 0.00 | 86.56 ± 0.19 | **87.29 ± 0.34** | 87.31 ± 0.25 | **87.46 ± 0.21** | 87.43 ± 0.07 |

Table A15: Delta between PaT and PaI test accuracy utilizing different pruning methods on ResNet18 with CIFAR-10. We mark with bold our proposed criteria.

| SPARSITY (%) | RANDOM | MAGNITUDE | GN | SNIP | GRASP | FD | FP | FTS | HD | HP | HTS |
|---|---|---|---|---|---|---|---|---|---|---|---|
| 10 | 0.76 | 0.66 | 0.72 | 0.57 | -0.27 | 0.50 | 0.73 | 0.81 | 0.84 | 0.77 | 0.83 |
| 20 | 0.81 | 0.84 | 0.92 | 0.86 | -1.79 | 0.91 | 0.72 | 0.98 | 0.73 | 0.90 | 1.17 |
| 30 | 0.98 | 0.44 | 0.79 | 0.72 | -2.48 | 0.86 | 0.89 | 0.84 | 1.10 | 1.03 | 1.03 |
| 40 | 0.92 | 1.14 | 0.78 | 0.62 | -2.93 | 0.84 | 0.84 | 0.90 | 0.98 | 0.71 | 1.05 |
| 50 | 0.75 | 0.78 | 1.00 | 1.01 | -1.70 | 0.93 | 0.76 | 0.94 | 0.70 | 1.01 | 1.04 |
| 60 | 0.96 | 0.88 | 0.87 | 0.89 | -1.59 | 0.81 | 0.84 | 0.77 | 1.15 | 1.08 | 1.03 |
| 70 | 0.57 | 0.98 | 1.13 | 0.96 | -2.30 | 0.93 | 0.73 | 1.17 | 1.11 | 1.18 | 1.16 |
| 80 | -0.12 | 0.86 | 1.41 | 1.28 | -2.71 | 1.37 | 0.85 | 1.08 | 0.95 | 1.44 | 1.32 |
| 90 | -0.06 | 1.77 | 1.35 | 1.41 | -1.85 | 1.74 | 1.45 | 1.08 | 1.58 | 1.93 | 1.63 |
| 95 | -0.01 | 1.86 | 1.34 | 1.69 | -2.36 | 2.28 | 1.59 | 1.67 | 2.26 | 1.85 | 2.11 |
| 98 | -0.15 | 3.20 | 0.45 | 1.92 | -2.75 | -25.33 | 1.94 | 1.79 | 2.33 | 1.56 | 2.39 |
| 99 | 0.94 | 13.39 | -36.92 | 2.64 | -2.29 | -72.13 | 2.82 | 2.44 | 2.26 | 1.60 | 2.95 |

Frankle et al. (Frankle et al., 2020) previously criticized PaI methods for consistently underperforming compared to magnitude PaT. However, their analysis did not assess how the PaI-designed criteria perform in the PaT setting. Table A14 shows that second-order-based criteria remain top performers across sparsities in the PaT setting. Table A15 presents the test accuracy delta between PaI and PaT settings. As we increase the sparsity, it is clear that the advantage of magnitude PaT comes at the expense of training a fully dense model and the required retraining. In the case of the HP metric, it gained only 1.6%, showing a close gap between the settings.

# I COMPARISON OF OUR CRITERIA WITH MAGNITUDE-BASED PRUNING

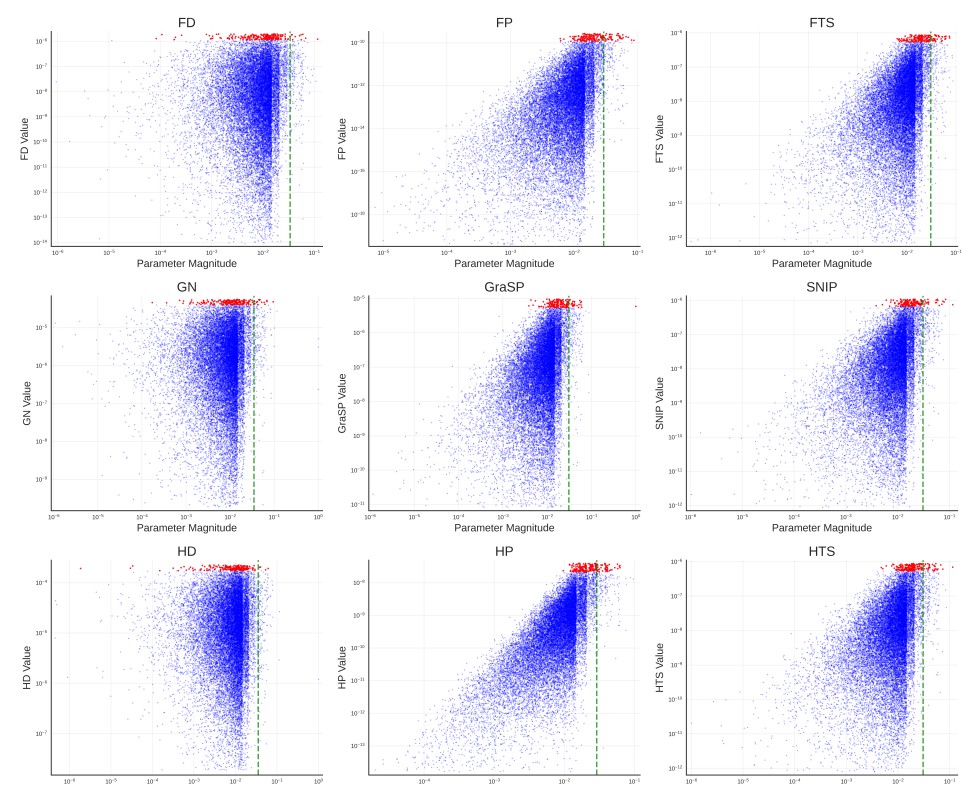

Figure A1: Proposed criteria vs. Magnitude parameter selection for 99% sparsity (ResNet18, CIFAR-10, Seed 0)

Figure A1 illustrates the relationship between parameter magnitude and different sensitivity-based pruning metrics. Each point represents a model parameter, with red points indicating the top-ranked parameters selected for retention by each criterion. The green dashed line marks the 99th percentile of parameter magnitudes. A key observation is that the success of data-based methods relies on the ability to consider small-magnitude parameters that would be discarded by magnitude-based pruning.

## J EMPIRICAL COMPARISON OF HESSIAN VERSUS FIM

We conducted simple experiments to evaluate the structural similarity between the Hessian matrix and the FIM. With a very small model (3 layers), we computed the two matrices per layer using two datasets: MNIST and CIFAR-10. We obtained the full Hessian using automatic differentiation and the FIM by accumulating the gradient's outer products. To compare them, we first normalized the matrices, which differ only in scale, as pointed out by a similar experiment in Singh & Alistarh (2020), and then compared the absolute difference between parameters: $|\text{Hessian} - \text{FIM}|$. We can identify that the majority of the structure is consistent between both estimations. The similarity becomes more noticeable after one training epoch, and it is consistent with the initial structure at initialization.

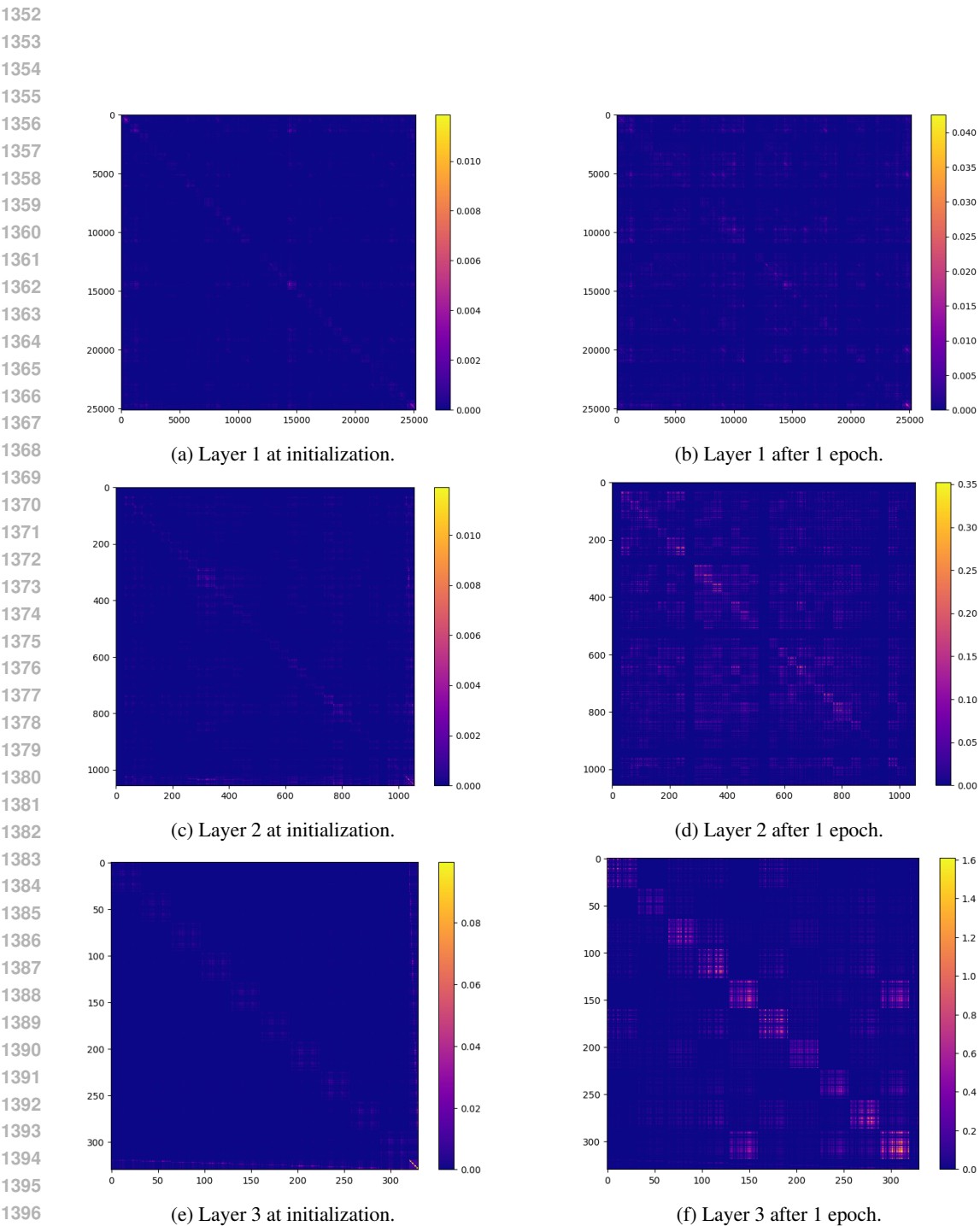

(a) Layer 1 at initialization.

(b) Layer 1 after 1 epoch.

(c) Layer 2 at initialization.

(d) Layer 2 after 1 epoch.

(e) Layer 3 at initialization.

(f) Layer 3 after 1 epoch.

Figure A2: Difference between normalized Hessian vs. FIM for MNIST dataset.

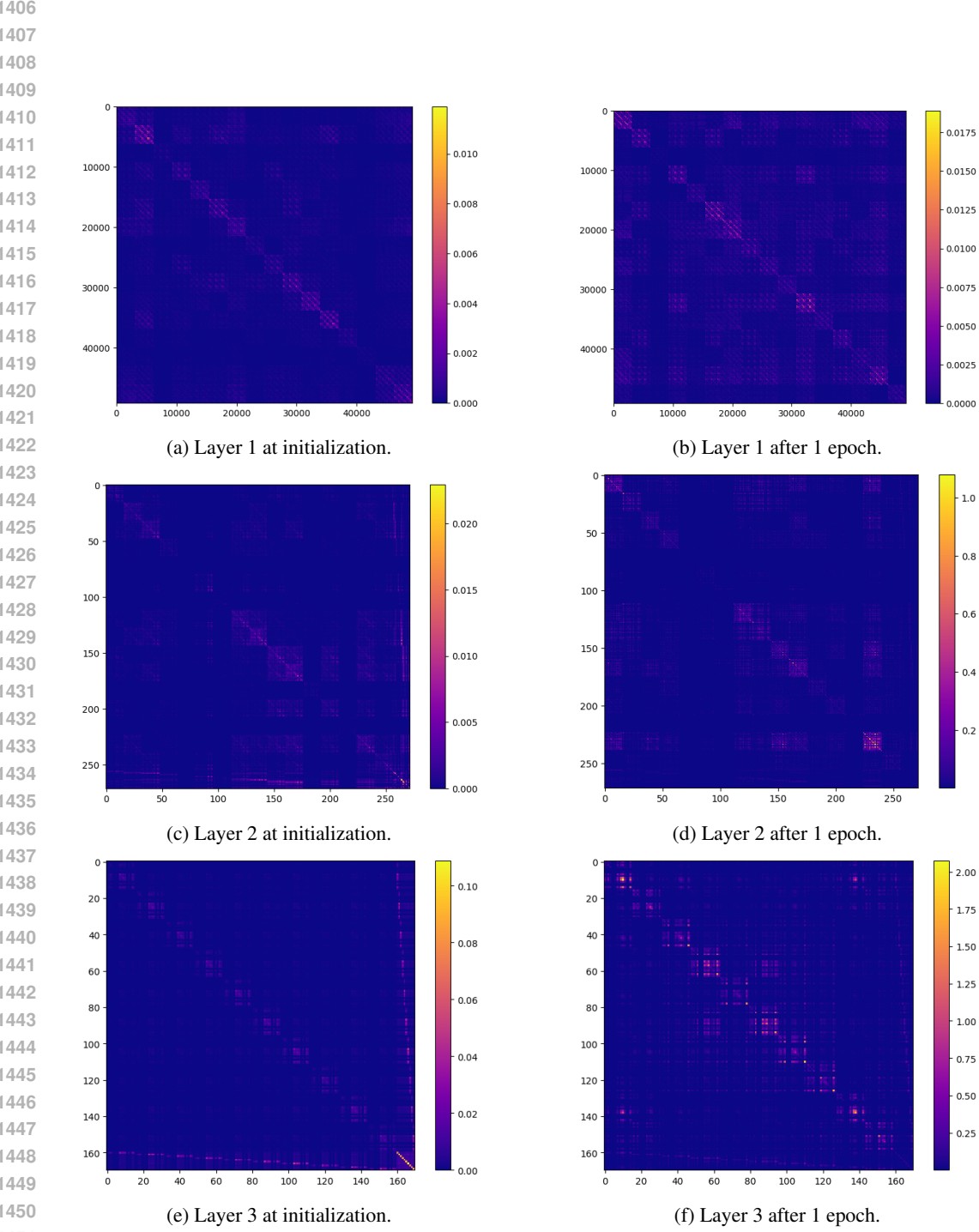

(a) Layer 1 at initialization.

(b) Layer 1 after 1 epoch.

(c) Layer 2 at initialization.

(d) Layer 2 after 1 epoch.

(e) Layer 3 at initialization.

(f) Layer 3 after 1 epoch.

Figure A3: Difference between normalized Hessian vs. FIM for CIFAR-10 dataset.

