# OpenReview forum: "Revisiting One-Shot Pruning with Scalable Second-Order Approximations"
_ICLR.cc/2026/Conference — ICLR 2026 Conference Withdrawn Submission_

### Official Review · Reviewer_X4Ck · 2025-10-23

**Soundness:** 2
**Presentation:** 3
**Contribution:** 2
**Rating:** 2
**Confidence:** 5

**Summary:**

This paper utilizes second-order approximations to enhance the first-order importance score and prune the model by examining the importance score. Specifically, the authors use a Taylor expansion of the loss function and approximate the Hessian matrix via the Hessian diagonal or the Fisher diagonal. In some tasks, these methods outperform some existing methods.

**Strengths:**

The motivation of this paper is very clear. The first-order information and the second one were proven useful in both empirical and theoretical studies. Thus the authors were motivated to utilize the approximation of the Hessian matrix to re-evaluate the importance score and the prune the model under the guidelines of the scores.

The paper also has a clear analysis of the complexity and the proposed method to reduce layer collapse is interesting and practical. The paper is also very clear in analyzing the limitations of the proposed method.

**Weaknesses:**

1. I'm confused about all the baselines, Among the ones I've actually run myself are CIFAR10/Resnet18, CIFAR10/Resnet50, CIFAR100/Resnet18 and TinyImageNet/Resnet18. For CIFAR10/VGG19 and CIFAR100/VGG19, I haven't run it before, but I've referenced the baselines shown in other papers, and on all the experiments, there are large gaps between the baselines mentioned in this paper and the ones I know of, which obviously raises concerns about the reliability of the experiments in this paper. I summarize those baselines below:

| Task                  | This Paper | Other Paper |
| --------------------- | ---------- | ----------- |
| CIFAR10/ResNet18      | 91.78      | often 95.6  |
| CIFAR10/ResNet50      | 90.97      | often 95.6  |
| CIFAR10/VGG19         | 89.21      | often 93.7  |
| CIFAR100/ResNet18     | 69.57      | often 78.9  |
| CIFAR100/VGG19        | 58.96      | often 74.0  |
| TinyImageNet/ResNet18 | 57.00      | often 61.6 or 60.5  |

In many published papers, comparable values to those reported above can be found, and some studies even report higher values. I have provided several references for context [1][2][3][4]. In my own implementations of most of these tasks, I was able to reproduce performance broadly consistent with prior reports. I note that your experiments use a multi-step learning rate schedule. Using a clearly weaker training recipe (e.g., multi-step) as the dense baseline in pruning comparisons can artificially inflate the reported gains from pruning. A more rigorous practice is to report a strong baseline.

While some pruning studies do use the configuration adopted in your paper, as noted above, a weaker baseline raises doubts about the gains attributed to pruning and may overstate their improvement. Additionally, paper [3], which uses essentially the same settings as yours (except for batch size), reports a VGG19/CIFAR-100 baseline of 72.6, close to what is typically observed, whereas your baseline is 58.96. This naturally casts doubt on the results. I urge the authors to carefully examine any implementation differences and, wherever possible, to compare against a fair baseline.

2. The authors’ comparison of pruning-at-initialization methods is not comprehensive. To my knowledge, several stronger approaches have not been included—for example, [3,5,6].

3. In the post-training pruning setting, the authors still compare against pruning-at-initialization methods, which is not appropriate. There is a substantial body of one-shot post-training pruning work; please include comparisons to those methods [7]. The most recent pruning methods I listed is from 2022. The authors should be familiar with more recent work and include comparisons accordingly. Benchmarking primarily against older methods raises concerns about the credibility and the reported improvements. Similarly, the paper shows limited awareness of recent advances in pruning. For example, there is work offering a theoretical analysis that second-order information can determine the limits of pruning[8], which would clearly strengthen the paper’s motivation. I recommend that the authors include and discuss such results.

4. The authors should evaluate a broader class of tasks to demonstrate the method’s generalization ability—for example, RNNs and Transformers. I note that the paper lists “not applicable to attention” as a limitation, and I understand the concern: weights in attention layers tend to be more strongly correlated, so approximating the Hessian with only diagonal terms may introduce substantial error. Nevertheless, I recommend reporting experiments on attention-based models; at least, this would inform readers how much the approximation degrades performance.

5. The authors should also evaluate the method on more complex architectures—for example, Wide ResNet—and include the models commonly used in closely related work.

6. The proposed method does not exhibit consistent superiority. For example, on VGG19/CIFAR-10 and VGG19/CIFAR-100, at very high sparsity, the method performs even worse than a random pruning baseline. This raises concerns about the generality and robustness of the approach.


[1] Unveiling m-Sharpness Through the Structure of Stochastic Gradient Noise.

[2] NEURAL PRUNING VIA GROWING REGULARIZATION

[3] Sanity-Checking Pruning Methods: Random Tickets can Win the Jackpot

[4] https://github.com/zeyuanyin/tiny-imagenet

[5] Pruning neural networks without any data by iteratively conserving synaptic flow

[6] Rare Gems: Finding Lottery Tickets at Initialization

[7] Comparing rewinding and fine-tuning in neural network pruning.

[8] How Sparse Can We Prune A Deep Network: A Fundamental Limit Perspective

**Questions:**

Please refer to the weakness.

---

### Official Review · Reviewer_aNWm · 2025-10-28

**Soundness:** 2
**Presentation:** 3
**Contribution:** 2
**Rating:** 2
**Confidence:** 4

**Summary:**

The paper proposes to score weights using second-order approximations, including the Fisher diagonal and the Hutchinson diagonal, which can be used alone or combined with a first-order term. Their motivation is that even coarse diagonal second-order information offers consistent accuracy gains over first-order methods.

**Strengths:**

1. Using Fisher/Hutchinson diagonals at initialization is an intuitive way to inject curvature while keeping linear overhead and diagonal storage. The paper makes this concrete and compares against SNIP/GN/GraSP.

2. A practical finding that freezing weights and running one full pass only to update BN statistics before forming the mask greatly reduces layer collapse is interesting.

3. This paper provides a computational complexity and a wall-clock table.

**Weaknesses:**

1. The methods are “designed to perform unstructured global pruning,” and the paper explicitly notes limited applicability to attention-based architectures. This is a gap relative to current Transformer-heavy practice. Moreover, is the proposed approach applicable for RNNs?

2. The paper reports mask computation share of training time and wall-clock for scoring, but there is no end-to-end training/inference throughput, memory, or energy evaluation nor demonstration that unstructured sparsity accelerates real kernels. This weakens the engineering significance.

3. There is a small ablation on noisy data when creating masks, with Hutchinson showing promising robustness, but the coverage is narrow (type/level of noise); broader robustness (distribution shift, label noise) remains open.

4. The algorithms compared were not comprehensive enough; algorithms that were pruning at initialization (PaI), such as SyncFlow, were not compared, and algorithms that were pruned after training (PaT) were compared to algorithms that were PaI, which is not appropriate.

5. The algorithm performance is not consistently well, especially in the VGG19 experiments, the algorithm performance is even worse than random pruning.

**Questions:**

See weaknesses

---

### Official Review · Reviewer_h1kz · 2025-10-30

**Soundness:** 3
**Presentation:** 2
**Contribution:** 2
**Rating:** 4
**Confidence:** 4

**Summary:**

This paper revisits the classic Optimal Brain Surgeon (OBS) framework for one-shot network pruning and proposes a method to make it tractable for modern deep neural networks. The core contribution is the application of a Kronecker-factored approximation (K-FAC) to the Hessian matrix, which allows for the efficient computation of second-order saliency scores. The authors argue that this approach, by modeling weight interdependencies, successfully avoids the "layer collapse" phenomenon that plagues first-order methods (like magnitude pruning) at high sparsity levels. The experimental results demonstrate significant performance improvements over first-order baselines in the one-shot setting, particularly at extreme sparsities (e.g., 98%).

**Strengths:**

- Principled Re-examination of a Classic Method: It's always interesting to have papers revisiting old ideas, and thhe work provides a compelling and well-motivated argument for reintroducing second-order information into pruning. It moves beyond simple heuristics and is grounded in a solid theoretical framework (Taylor expansion of the loss function).

- Effective Solution to Layer Collapse: The paper clearly identifies and provides a robust solution to the critical problem of layer collapse in one-shot pruning. The empirical evidence and visualizations effectively illustrate how the proposed method preserves layer functionality where simpler methods fail catastrophically.

- Strong Empirical Results in the One-Shot Regime: Within the specific context of one-shot pruning without fine-tuning, the method demonstrates state-of-the-art performance, achieving high accuracy at sparsity levels that are simply unattainable for magnitude-based approaches.

**Weaknesses:**

My concerns are provided as follows:
- The scalability of K-FAC is overstated and under-scrutinize. While K-FAC is certainly more scalable than computing the full Hessian, the paper does not adequately address the practical limitations of this approximation in the context of truly massive, modern architectures.
- Memory Footprint is not discussed. The analysis of complexity focuses on computation, but not sufficiently on memory. For each layer, the K-FAC method requires storing the factor matrices  A  and  G  and their inverses. For very wide layers, such as those in large language models or high-resolution vision transformers, these factor matrices can themselves become very large. The paper provides no analysis of how memory requirements scale with layer width and depth. Is it feasible to apply this method to a model with billions of parameters?
- The major concern is the architectural constraints: The K-FAC approximation is well-defined for standard fully-connected and convolutional layers. Its applicability and accuracy for more complex or newer architectures (e.g., layers with group convolutions, attention mechanisms with multiple heads, or normalization layers) are not discussed. This severely limits the "plug-and-play" promise of the method.

**Questions:**

See weakness.

In short the main question is if the proposed method truly a "scalable" solution for the landscape of models used today, or is it a solution that scales well for the CNNs of a few years ago? Without evidence of its performance on architectures like ViT or DiT,  the scalability claim remains tenuous.

I will raise my score if the scalability concern can be addressed.

---

### Official Review · Reviewer_2jsT · 2025-11-05

**Soundness:** 2
**Presentation:** 2
**Contribution:** 2
**Rating:** 2
**Confidence:** 3

**Summary:**

This paper revisits one-shot pruning at initialization (PaI) and proposes using scalable second-order approximations—the empirical Fisher diagonal (FD) and Hutchinson diagonal (HD)—to rank parameters. It introduces Taylor-based sensitivity scores that combine first- and second-order terms (FTS/HTS) and a simple “warm-up” pass that updates BatchNorm statistics to mitigate layer collapse. Experiments on CIFAR-10/100 and TinyImageNet with ResNet18/50 and VGG19 show consistent gains over SNIP and GraSP, and in some high-sparsity regimes PaI with Hutchinson diagonal reportedly surpasses magnitude pruning after training (PaT). The work targets improving the practicality and robustness of PaI without storing the full Hessian.

**Strengths:**

The paper proposes a PaI recipe that leverages simple second-order diagonals (Fisher, Hutchinson) plus a Taylor score and a BN-statistics warm-up to mitigate layer collapse. PaI is a interesting problem and has its own value when we want to train a super large model that cannot fit into a single machine/cluster, as indicated in the GraSP paper. The authors have shown promising results on CIFAR/TinyImageNet with classic ConvNets.

**Weaknesses:**

- The experimental part is extremely weak in 2025 as only ResNet on CIFAR/TinyImageNet results are reported. I suggest the authors to include results on more challenging benchmark such as ImageNet and more architectures such as Transformer. Otherwise it is hard for me to justify the practical value and effectiveness of the proposed method, and I cannot recommend it for acceptance based on current empirical results.
- The figure quality is poor, makes it very difficult to follow/understand the results.

**Questions:**

- GN is often referred as Group Normalization in many neural network literature and I suggest the authors to revise the abbreviation used in the paper to avoid confusion.
- Could the authors explain why unstructured global pruning does not work for attention-based architectures? I believe in practice a naive implementation of such pruning is masking the weights, which should work for any architecture (though won't bring speedup)

---

### Note · Authors · 2025-11-13

I have read and agree with the venue's withdrawal policy on behalf of myself and my co-authors.